



# Performance of the Adriatic Sea and Coast (AdriSC) climate component – a COAWST V3.3-based coupled atmosphere-ocean modelling suite: ocean part

Petra Pranić[1], Cléa Denamiel[1,2], Ivica Vilibić[1]

[1]Institute of Oceanography and Fisheries, Šetalište I. Meštrovića 63, 21000 Split, Croatia
[2]Ruđer Bošković Institute, Division for Marine and Environmental Research, Bijenička cesta 54, 10000 Zagreb, Croatia

*Correspondence to*: Petra Pranić (pranic@izor.hr)

**Abstract**

In this study, the Adriatic Sea and Coast (AdriSC) kilometre-scale atmosphere-ocean climate model covering the Adriatic and northern Ionian Seas is presented. The AdriSC ocean results of a 31-year long (i.e. 1987-2017) climate simulation, derived with the Regional Ocean Modeling System (ROMS) 3-km and 1-km models, are evaluated with respect to a comprehensive collection of remote-sensing and *in situ* observational data. In general, it is found that the AdriSC model is capable to reproduce the observed sea-surface properties, daily temperatures and salinities and the hourly ocean currents with good accuracy. In particular, the AdriSC ROMS 3-km model demonstrates skill in reproducing the main variabilities of the sea-surface height as well as the sea-surface temperature, despite a persistent negative bias within the Adriatic Sea. Furthermore, the AdriSC ROMS 1-km model is found to be more capable to reproduce the observed thermohaline and dynamical properties than the AdriSC ROMS 3-km model. For the temperature and salinity, better results are obtained in the deeper parts than in the shallow shelf and coastal parts, particularly for the surface layer of the Adriatic Sea. The AdriSC ROMS 1-km model is also found to perform well in reproducing the seasonal thermohaline properties of the water masses over the entire Adriatic-Ionian domain. The evaluation of the modelled ocean currents revealed better results at locations along the eastern coast and especially the north-eastern shelf than in the middle-eastern coastal area and the deepest part of the Adriatic Sea. Finally, the AdriSC climate component is found to be a more suitable modelling framework to study the dense water formation and long-term thermohaline circulation of the Adriatic-Ionian basin than the available Mediterranean regional climate models.

## 1 Introduction

Due to the temporal and spatial sparsity of the *in situ* observations, the study of the dynamics and variability of the ocean processes mostly relies on the constant developments and improvements of the available numerical modelling tools. Over the years, in the Adriatic Sea, significant progresses have thus been made by the ocean modelling community to overcome the challenges posed by the complex geomorphology of the region (Figs. 1.a and 1.b): (1) an extremely complex coastline with





over 1200 islands along the eastern coast, (2) bathymetries ranging from a shallow shelf (30 m on average) in the north to a very deep pit (up to approximately 1200 m) in the south and (3) mountain ranges – Alps in the North, Apennines in the West and Dinarides in the East – surrounding the semi-enclosed elongated Adriatic basin.

Historically, numerous studies have focused on the numerical modelling of the dense water formation and spreading in the
Adriatic due to its vital role for many ocean processes such as the Adriatic-Ionian thermohaline circulation (Orlić et al., 2006; Vilibić et al., 2013), the Adriatic-Ionian Bimodal Oscillation System (hereafter referred as BiOS; Gačić et al., 2010; Gačić et al, 2014) as well as the biogeochemical properties of the ocean (Gačić et al., 2002; Krasakopoulou et al., 2005; Boldrin et al., 2009; Batistić et al., 2014). The initial numerical efforts in studying dense water formation in the Adriatic, with ocean model resolutions up to 3-km, were mainly focused on the North Adriatic Dense Water (NadDW) formation within the northern
Adriatic shelf (Bergamasco et al., 1999; Beg-Paklar et al., 2001) as well as the Adriatic Deep Water (AdDW) formation within the Southern Adriatic Pit and its interannual variability (Mantziafou and Lascaratos, 2004; 2008). At the time, the atmospheric fields used to force the ocean models were mostly climatological data (May, 1982; Artegiani et al., 1997) or the ECMWF (European Center for Medium-Range Weather Forecasts) global datasets (e.g. ERA 40, ERA-I; Vested et al., 1998; Zavatarelli et al., 2002; Oddo et al., 2011). However, many studies have demonstrated that the ECMWF reanalyses, due to their spatial
homogeneity and coarse resolution, could not properly reproduce the extreme bora events driving the dense water formation in the northern Adriatic Sea. In particular, Cavaleri and Bertoti (1997) highlighted that the underestimation of the bora wind speed could reach up to 50% which consequently led to a strong underestimation of NAdDW production rates (Vilibić and Supić, 2005). Therefore, in some studies, the ECMWF wind speeds have been increased by up to 20% in order to improve the representation of the ocean dynamics during bora events (e.g. Mantziafou and Lascaratos, 2004). More recently, the
implementation by Janeković et al. (2014) of a modelling system based on the Regional Ocean Modeling System (ROMS; Shchepetkin & McWilliams, 2009) at 2 km of resolution forced by the operational atmospheric model ALADIN/HR (Aire Limitée Adaptation Dynamique développement InterNational; Tudor et al., 2013) has allowed for a better representation of the atmosphere-ocean dynamics during bora events in the northern Adriatic (Vilibić et al., 2016; Mihanović et al., 2018; Vilibić et al., 2018), which is also substantially influenced by the ocean feedback to the atmosphere (Pullen et al., 2006, 2007; Ličer
et al., 2016).

In the last decade, other studies have also used kilometer-scale limited-area models to simulate ocean processes driven by extreme conditions in the Adriatic Sea including, for example, extreme waves and storm surges as well as sea surface cooling, water column mixing, dense water formation and long-term thermohaline circulation occurring during severe bora and sirocco windstorms (e.g. Cavaleri et al., 2010, 2018; Ricchi et al., 2016; Denamiel et al., 2020a). However, aside from the atmospheric
forcing, other sources of errors have been documented to influence the quality of the Adriatic numerical modelling such as the representation of the river discharges and the choice of the open boundary conditions. Concerning the problems associated with the river discharges, the use of old river climatologies (Raicich, 1994) as well as the lack of recent river load observations,



in particular along the eastern Adriatic coast, resulted in large overestimation (multiplied by 5 at least) of the discharges in the north-eastern Adriatic (Janeković et al., 2014). However, these old climatologies have been used in many recent Adriatic

modelling studies (e.g. Zavatarelli and Pinardi, 2003; Oddo et al., 2005; Benetazzo et al., 2014), despite being proved to prevent the dense water generation in the coastal eastern Adriatic area (Mihanović et al., 2013) as well as to decrease the ocean density for up to 0.5 kg/m$^3$ in the primarily dense water formation sites in the northern Adriatic shelf (Vilibić et al., 2016). Concerning the propagation of errors from the open boundaries (particularly at the Strait of Otranto), they have mostly been documented as an underestimation of the salinity also linked to wrong freshwater forcing (Janeković et al., 2014). Other sources of errors,

like improper parameterization of vertical mixing and diffusion can also affect the performances of the Adriatic models and better ocean modelling solutions can be reached through a data assimilation procedure (Janeković et al., 2020). Notwithstanding the corrections of the above-mentioned sources of error, a common key conclusion of all the recent Adriatic studies was still the need for higher resolution atmospheric models and longer-term simulations to capture the coastal ocean dynamics in the Adriatic Sea.

In terms of long-term climate modelling, the Adriatic Sea has, till now, dominantly been studied with Regional Climate Models (RCMs) developed over the entire Mediterranean Sea within the Med-CORDEX initiative (e.g. Somot et al., 2006; Sevault et al, 2014). However, these RCMs have shown to be incapable to properly reproduce the processes at the coastal scale mainly due to their relatively coarse horizontal resolution (of the order of 10 km) which is insufficient to resolve the complexity of the coastal morphologies of the Adriatic (McKiver et al, 2016; Dunić et al., 2019). In addition, some quasi-climate ocean

studies were carried out to quantify interannual variability of the Adriatic dense water dynamics (e.g. Mantziafou and Lascaratos, 2004, 2008), while quantification of the sources for the Adriatic-Ionian decadal thermohaline variability required multi-decadal climate simulations of the Eastern Mediterranean (Theocharis et al., 2014).

Therefore, to quantify the impacts of climate change in the Adriatic, it is crucial to obtain an adequate representation at climate scales of the atmosphere-ocean interactions during extreme events which are, for example, driving the formation of dense

water within the basin. Atmospheric RCMs generally fail to provide such a representation, especially in the northern Adriatic where they cannot be used to study the extreme bora dynamics (Denamiel et al., 2020b, 2021a). Additionally, it has also been recently demonstrated that the latest higher resolution ECMWF reanalysis dataset – the ERA5 product (Hersbach et al., 2018), cannot be used either as a reference for climate model evaluation nor as a forcing for ocean models during bora events in the northern Adriatic as it also strongly underestimates the extreme bora speeds (Denamiel et al., 2021a). Consequently, in the

recent study of Liu et al. (2021) – which investigated the BiOS variability using a regional ocean circulation model at 9-km resolution driven by the ERA-20C atmospheric forcing (Poli et al. 2016) with a 101-year long simulation – the impact of the bora events not properly represented by the atmospheric forcing was mimicked by artificially setting up the 2-m air temperature and the dew point temperature over the entire Adriatic Sea to 0 °C in January, February, November and December.



Following the findings of the previous research, a need for higher resolution atmospheric models – which are capable to
reproduce the wind dynamics and air-sea interactions in the northern Adriatic – has been raised. Nevertheless, the development
of high-resolution atmosphere-ocean models in areas of the Mediterranean which are inadequately represented by regional
climate models, is still not in the focus of the Med-CORDEX climate community, mainly because of their extremely high
computational costs (Prein et al., 2015). The high-resolution atmosphere-ocean Adriatic Sea and Coast (AdriSC) climate model
(up to 3-km in the atmosphere and 1-km in the ocean) was thus implemented and a 31-year long evaluation simulation was
performed for the 1987-2017 period. In this work, the performance of the AdriSC ocean coastal model is evaluated while the
skill assessment of the AdriSC atmospheric kilometre-scale model is done in a separate study (Denamiel et al., 2021b). In
general, a proper evaluation of a high-resolution climate model is not a trivial task and most often the biggest challenge turns
out to be the availability, incompleteness and scarcity of observational data, as well as the imperfections of the observing
systems which set further limits on the evaluation process (Horak et al., 2021). More specifically, the evaluation of the ocean
climate models is known to be particularly challenging due to the sparsity and inhomogeneity in time of the ocean observations
(Somot et al., 2018). Additionally, the absence of standardized gridded products in the Adriatic Sea renders the inter-
comparison of the skills of such ocean climate models extremely difficult. To overcome these challenges, a significant effort
has been made in this study to collect, from various sources and institutions, a large number of historical observational ocean
data, especially long-term records and products with high temporal resolution and spatial coverage.

In the following section, the AdriSC climate component and the set-up of the AdriSC ocean model as well as the observations
and methods used to perform the skill assessment of the model are presented first. Then, in Section 3, the main results of the
study are presented and discussed in detail. They consist in three different kind of evaluations: (1) sea-surface (sea-surface
height and temperature) properties, (2) thermohaline properties (temperature and salinity) and (3) dynamical properties (current
speed and direction). Lastly, the findings of this study are summarized in Section 4.

## 2 Model, data and methods

### 2.1 AdriSC climate model

The Adriatic Sea and Coast (AdriSC; Denamiel et al., 2019) climate component is built around a modified version of the
Coupled Ocean-Atmosphere-Wave-Sediment-Transport (COAWST V3.3) modelling system (Warner et al., 2010) in order to
provide kilometre-scale hourly results for 31-year long simulations as described in Denamiel et al. (2021b). In this study, the
ocean results of the evaluation run for the 1987-2017 period – which can be easily accessed and retrieved via the web interface
https://vrtlac.izor.hr/ords/adrisc/interface_form (Ivanković et al., 2019; Denamiel et al., 2021b) – are presented in detail while
the set-up of the AdriSC climate model is summarized in Table 1. Hereafter, the Adriatic atmospheric processes are simulated
with the Weather Research and Forecasting (WRF v3.9.1.1) model (Skamarock et al., 2005) for a 3-km grid covering the entire
Adriatic and northern Ionian Sea (Fig. 1.a). Concerning the ocean, the Regional Ocean Modeling System (ROMS svn 885;



Shchepetkin & McWilliams, 2009) reproduces (1) the Adriatic-Ionian exchanges with a 3-km grid (266 x 361) identical to the atmospheric domain (Fig. 1.a) and (2) the complex coastal Adriatic Sea dynamics with a nested 1-km grid (676 x 730). Finally, the data exchanges between the WFR 3-km atmospheric grid and the ROMS 3-km & 1-km ocean grids are achieved with the Model Coupling Toolkit (MCT v2.6.0; Larson et al., 2005) and the remapping weights are computed with the Spherical Coordinate Remapping and Interpolation Package (SCRIP).

As described in Denamiel et al. (2021b), the COAWST model is compiled with the Intel 17.0.3.053 compiler, the PNetCDF 1.8.0 library and the MPI library (mpich 7.5.3) on the European Centre for Middle-range Forecast's (ECMWF's) High Performance Computing Facility (HPCF). Furthermore, the ecFlow work flow package used by all ECMWF operational suites is set-up to automatically and efficiently run the AdriSC long-term simulations in a controlled environment. Regarding the workload, the AdriSC climate model optimally runs on 260 CPUs with both the WRF and ROMS grids decomposed in 10 x
13 tiles and without hyper-threading.

[Table 1]

For a complete presentation of the AdriSC climate component, a detailed description of the set-up of both atmospheric and oceanic models is necessary. Since the evaluation of the AdriSC climate model is done separately for the atmosphere (Denamiel et al., 2021b) and the ocean, only the set-up of the AdriSC ROMS 3-km and ROMS 1-km models is described below.

First, a Digital Terrain Model (DTM) including: (1) coastline data generated by the Institute of Oceanography and Fisheries, (2) offshore bathymetry from ETOPO1 (Amante and Eakins, 2009) and (3) nearshore bathymetry from navigation charts CM93 201, is providing the high-resolution bathymetry data for both AdriSC ROMS grids. Moreover, the bathymetry (with the minimum depth of 2 m) is smoothed with the application of a Linear Programming (LP) method (Dutour Sikiric et al., 2009) to the ROMS 3-km and 1-km grids. In this way the roughness factors are minimized while keeping the DTM bathymetric
features. Also, the horizontal pressure gradient errors generated by the use of terrain-following coordinates with steep bathymetric gradients, are reduced. In the actual configuration of the AdriSC ROMS climate models, 35 vertical layers – transformed ( $Vtransform = 2$ ) and stretched ( $Vstretching = 4$ ) following Shchepetkin (2009) – are used with increased resolution at the surface ( $\theta_s = 6$ ) and bottom ( $\theta_b = 2$ ) as well as a thickness of 50 m ( $h_c = 50$ ).

Second, regarding the external forcing of the AdriSC ROMS 3-km model, the initial conditions and boundary forcing –
including sea-surface height, barotropic and baroclinic currents as well as baroclinic temperature and salinity – are provided daily by the Mediterranean Forecasting System (MFS) MEDSEA v4.1 re-analysis (resolution of 1/16° x 1/16°; Pinardi et al., 2003) distributed by the Copernicus Marine Environment Monitoring Service (CMEMS). The tidal forcing consists in 8 tidal constituents (M2, S2, N2, K2, K1, O1, P1, Q1) extracted from the Mediterranean and Black Seas (2011) 1/30° regional solution of the OSU Tidal Inversion Software (OTIS; Egbert et al., 1994; Egbert and Erofeeva, 2002). The used tidal constituents were





previously found to adequately reproduce the tidal dynamics in the Adriatic Sea (Cushman-Roisin and Naimie, 2002; Janeković and Kuzmić, 2005). Concerning the river forcing, 54 river flows in total (only 49 for the 1-km grid) are imposed over at least 6 grid points each (and 18 grid points for the Po river delta), with river mouths located along the coastline of: Italian peninsula, Sicily, Croatia, Slovenia, Albania, Montenegro and Greece. The monthly climatology of the river flow is acquired from the RivDis database (Vörösmarty et al., 1996), and studies from Pano and Abdyli (2002), Malačič and Petelin

(2009), Pano et al. (2010), Janeković et al. (2014) and Ljubenkov (2015), whereas the river flow interannual variability is obtained from Ludwig et al. (2009). Additionally, the river flows are vertically distributed between the 20 first sigma levels.

Third, on the one hand, the high optical water clarity in shallow parts of the Adriatic such as the eastern Adriatic Sea creates a warming sea-surface temperature (SST) trend linked to the absorption of the shortwave radiation reaching the seafloor while, on the other hand, the low optical water clarity along the Italian coast due the muddy waters of the Po river plume tends to

produce opposite trends. A dQ/dSST procedure, described in detail in the study of Denamiel et al. (2019), is thus used to solve this problem by minimizing the corrections of the heat fluxes produced by WRF, while making sure that no artificial SST trends are generated in the shallow parts of the ROMS grids. In brief, this method imposes a heat flux correction through the calculation of the kinematic surface net heat flux sensitivity to the SST of reference. Consequently, the 9-km SST forcing from the MEDSEA v4.1 re-analysis is also used as reference for the calculation of the dQ/dSST procedure with the ROMS model.

Finally, concerning the configuration of the physical options for the ROMS models, the MEDSEA barotropic velocities, surface elevations and baroclinic fields at the open boundaries are imposed with the Flather (Flather, 1976), Chapman (Chapman, 1985) and Orlanski (Orlanski, 1976) conditions. Additionally, the baroclinic structure is relaxed – with a minimum folding time of 3 days – towards the fields provided by the MEDSEA ocean climatology (Marchesiello et al., 2001). The relaxation occurs in two different nudging areas: (1) a ten grid point wide zone along the open boundaries, and (2) a zone

covering the bathymetry deeper than 2000 m but only for the temperature and salinity, in order to minimize the numerical diapycnal mixing. A sponge area of ten grid points (identical to the first nudging area) also insures that the horizontal viscosities are smoothly interpolated from values four times bigger at the open boundaries than inside of the domain. Last, the tracer advection is provided with the Multidimensional Positive Definite Advection Transport Algorithm (MPDATA; Smolarkiewicz and Grabowski, 1990) while the horizontal momentum advection uses a fourth-order cantered scheme and the turbulence

closure scheme follow the GLS gen framework (Umlauf and Burchard, 2003).

## 2.2 Skill assessment

### 2.2.1 Observations

In this study, the AdriSC ocean model (ROMS 3-km and ROMS 1-km) performances are assessed for 5 different variables (sea-surface height, temperature, salinity, ocean current speed and direction) by comparison to a comprehensive collection of

observational data retrieved for the 1987-2017 period from *in situ* measurements and remote-sensing gridded products.



The first dataset used in this study is the Sea Surface Height Anomalies (SSHA) gap-free remote sensing (L4) product, SEA_SURFACE_HEIGHT_ALT_GRIDS_L4_2SATS_5DAY_6THDEG_V_JPL1812 (Zlotnicki et al., 2019; hereafter referred as JPL MEASURES). It is produced at the Jet Propulsion Laboratory (JPL) of the Physical Oceanography Distributed Active Archive Center (PODAAC) on a 1/6° grid every 5 days since October 1992. The final gridded product, obtained by a

kriging method, is a combination of SSHA data derived from TOPEX/Poseidon, Jason-1, Jason-2 and Jason-3 as reference data as well as ERS-1, ERS-2, Envisat, SARAL-AltiKa, CRyosat-2, depending on the date.

Second, two different sea-surface temperature (SST) gap-free remote sensing (L4) products were chosen for this evaluation. They were extracted from the datasets provided by the Group for High Resolution Sea Surface Temperature (GHRSST) which offers a framework for SST data sharing and processing. The first product, AVHRR_OI-NCEI-L4-GLOB-v2.0 (National

Centers for Environmental Information, 2016; hereafter referred as AVHRR), is produced daily, on a 0.25° grid, at the National Oceanographic and Atmospheric Administration (NOAA) National Centers for Environmental Information (NCEI) since September 1981. It uses Optimal Interpolation (OI) by interpolating and extrapolating SST observations from the Advanced Very High-Resolution Radiometer (AVHRR) and *in situ* platforms (i.e. ships and buoys), resulting in a smoothed complete field. The main advantage of AVHRR is thus that it covers the entire 1987-2017 period of the AdriSC evaluation run. However,

its resolution is rather coarse and is likely to be insufficient to properly describe the coastal areas of the Adriatic basin. Consequently, a high resolution second product is also used in this study for a shorter period. The MUR-JPL-L4-GLOB-v4.1 (JPL MUR MEaSUREs Project, 2015; hereafter referred as JPL MUR), is indeed produced daily, on a global 0.01° grid, at the JPL of the PODAAC since June 2002. It uses wavelets as basis functions in an OI approach. The Multiscale Ultrahigh Resolution (MUR) analysis is based upon night time GHRSST SST observations from several instruments including the

National Aeronautics and Space Administration (NASA) Advanced Microwave Scanning Radiometer-EOS (AMSR-E), the JAXA Advanced Microwave Scanning Radiometer 2 on GCOM-W1, the Moderate Resolution Imaging Spectroradiometers (MODIS) on the NASA Aqua and Terra platforms, the US Navy microwave WindSat radiometer, the AVHRR on several NOAA satellites, and *in situ* SST observations from the NOAA iQuam project.

The third dataset consists in a comprehensive collection of temperature and salinity *in situ* Conductivity Temperature Depth

(CTD) observations with diverse temporal and spatial coverages (Fig. 1b). This dataset combines 17 different experiments and/or scientific cruises: (1) Argo floats – ARGO (https://argo.ucsd.edu), (2) ASCOP project Phase 2, Istituto Nazionale di Oceanografia e di Geofisica Sperimentale (OGS), (3) Corfu System Project – CSP01 cruise (https://isramar.ocean.org.il/PERSEUS_Data), (4) Dynamics of the Adriatic in Real Time – DART_CTD (Martin et al., 2009; Burrage at al., 2009), (5) CTD observations, Institute of Oceanography and Fisheries (IOR) – IOR_Data_CTD, (6) Palagruža

transect long-term observations – IOR_Pal_CTD, (7) Mediterranean Data Archaeology and Rescue project – MEDATLAS (http://www.ifremer.fr/medar/cdrom_database.htm), (8) Northern Adriatic Experiment CTD observations – NAdEx_CTD (Vilibić et al., 2018), (9) Otranto Gap Experiment, SACLANT Undersea Research Centre – OTRANTO, (10) PALMAS, OGS,





(11) PCO, Consiglio Nazionale delle Ricerche (CNR), Istituto di Biologia del Mare, Venice, (12) Physical Oceanography of the Eastern Mediterranean project, Hellenic National Oceanographic Data Centre (HCMR/HNODC) – POEM, (13)

Programma di RIcerca e Sperimentazione del Mare Adriatico Phase 2 (chemical stations) hosted at OGS – PR2_UR, (14) Programma di RIcerca e Sperimentazione del Mare Adriatico Phase 1 hosted at OGS – PRISMA, (15) PRV, CNR, Istituto di Biologia del Mare, Venice, (16) Northern Adriatic long-term observations, Ruđer Bošković institute – RB_NAd (Vilibić et al., 2019), (17) SIRIAD cruise hosted at OGS – SIRIAD_15. This large dataset includes over 7000 locations in total and covers almost entirely the Adriatic Sea and partially the northern Ionian Sea. Data sampling frequency varies largely depending on

the locations and the observations while the maximum depth of the measurements ranges between 40 and 2140 m. All CTD observations had been already independently quality checked except IOR_DATA_CTD, which was in this study quality controlled to automatically and visually remove outliers, values with steep gradients and vertical instabilities using standard procedures described in the SeaDataNet manual (https://www.seadatanet.org/Standards/Data-Quality-Control).

Finally, the last dataset is a collection of ocean currents speed and direction combining Acoustic Doppler Current Profiler

(ADCP) and Rotor Current Meter (RCM) *in situ* observations with diverse temporal coverage (Fig. 1c). This dataset combines 7 different experiments and/or scientific cruises: (1) Dynamics of the Adriatic in Real Time – DART_ADCP (Martin et al., 2009; Burrage at al., 2009), (2) East Adriatic Coastal Experiment – EACE (http://www.izor.hr/eace/eace_g.htm), (3) historical RCM observations – IOR_Data_RCM, (4) Palagruža transect ADCP observations following the winter of 2012 – IOR_Pal_ADCP, (5) Jadranski projekt Phase 1 ADCP observations – JP1, (6) Jadranski projekt Phase 2 ADCP observations – JP2, (7) Northern

Adriatic Experiment ADCP observations – NAdEx_ADCP (Vilibić et al., 2018). All the observations had been already independently quality checked except for IOR_Data_RCM, which received an additional QC performed to automatically and visually remove obvious outliers, spurious data and long strings of constant values. A full list of the data collected to perform the AdriSC ROMS 3-km and 1-km model evaluations during the 1987-2017 period is presented in Table 2. The table includes, for each of the four datasets (i.e. SSHA, SST, CTD and ADCP/RCM), the name of the corresponding observations (i.e. remote

sensing products as well as CTD and ADCP/RCM experiments and/or scientific cruises), the time period, the number of locations, the number of records and the maximum measured depth.

[Table 2]

### 2.2.2 Methods

Once the evaluation run is completed, the extraction of the AdriSC ROMS 3-km and ROMS 1-km model hourly and daily

results is achieved in two different ways. A bilinear interpolation to the coarser coordinates of the JPL MEASURES, AVHRR and JPL MUR gridded products with the Earth System Modelling Framework (ESMF) software is performed for the comparison with satellite observations. While, for the comparison with the *in situ* observational datasets, a near-neighbour method at points in time and space matching the coordinates of the CTD, RCM and ADCP stations is used before linear





interpolation to the vertical structure of the measurements following the depth. Moreover, in order to obtain more robust

statistics for the chosen geophysical quantities which are likely to be heavy tailed due to extreme conditions, the use of median and Median Absolute Deviation (MAD) is preferred to the mean and standard deviation preconized for normal distributions. Finally, the performance of the AdriSC ocean models is evaluated separately for each type of observational dataset (i.e. SSHA and SST remote sensing gridded products, CTD observations and ADCP/RCM measurements).

Due to the relatively coarse temporal and spatial resolutions of the satellite observations, only the evaluation of the AdriSC

ROMS 3-km model is performed against the selected sea-level and sea-surface temperature remote sensing products (i.e. SSHA from JPL MEASURES and SST from AVHRR and JPL MUR). For the sea-level analysis, Empirical Orthogonal Functions (EOFs) are used to compare, in space and time, the most important variability patterns in the Adriatic and northern Ionian seas. Indeed, Gačić et al. (2011) have demonstrated that the BiOS – consisting in the decadal switch from cyclonic to anti-cyclonic of the circulation in the northern Ionian Sea and greatly impacting the thermohaline circulation of the Adriatic Sea – is well

described with the decadal change of sign of one of the main components of the EOF derived from SSHA products. The EOFs (also known as Principal Component Analysis or Eigen Analysis) presented in this study are obtained via a covariance matrix and are normalized (i.e. the sum of squares for each EOF pattern equals one). The time series of the amplitudes (also known as principal components or expansion coefficients) associated with each eigenvalue in the EOF are derived via the dot product of the data and the EOF spatial patterns and the mean is subtracted from the value of each component time series. Consequently,

EOFs performed on SSHA from remote sensing products and Sea-Surface Height (hereafter referred as SSH) results from the AdriSC ROMS 3-km model can be directly compared despite the different mean sea-level references used to derive SSHA and SSH. For the SST analysis, the bias or difference between model results and observations is calculated at each point in time and space of the AVHRR and JPL MUR datasets. The biases are then analysed in space with statistical quantities such as median and 1st, 25th, 75th and 99th percentiles.

The skill assessment of the AdriSC ROMS 3-km and 1-km models against *in situ* temperature and salinity CTD data is divided into three main analyses. First, a basic assessment of the model performances is achieved with (1) Taylor diagrams (Taylor, 2001) using multiple statistical parameters, (2) Quantile-Quantile (Q-Q) plots comparing the distributions of the observed and modelled temperature and salinity and (3) scatter diagrams uniquely for the AdriSC ROMS 1-km model, which is further used in the study as having a more precise matching of the nearest grid points with the CTD locations. The second analysis looks

in more detail at the spatial distributions of the median and MAD of the biases between the AdriSC ROMS 1-km model and the CTD observations depending on the depth (i.e. for 4 different depth ranges: 0-50 m, 50-200 m, 200-500 m and 500-2000 m). Finally, a climatological analysis of the AdriSC ROMS 1-km results is performed for 7 different subdomains: Western Coast, Northern Adriatic, Middle Adriatic, Kvarner Bay, Deep Adriatic, Dalmatian Islands and Otranto-Ionian (Fig. 1b). For each subdomain, the following results are presented: (1) monthly climatology of the median (and MAD as upper and lower

bounds) of the modelled and observed temperature and salinity – this analysis is done without taking the same depth range for each month due to the vertical scar

sity of the measurements, (2) seasonal variations of the vertical profiles of median temperature and salinity biases interpolated to standard oceanographic depths selected appropriately for each subdomain – seasons are defined as January February March (JFM) for winter, August May June (AMJ) for spring, July August September (JAS) for summer and October November

December (OND) for autumn – and (3) seasonal variations of Temperature-Salinity (T-S) diagrams of observations and model results.

Lastly, the evaluation of the AdriSC ROMS 3-km and 1-km models against *in situ* ocean current speed and direction of the ADCP and RCM measurements is divided into two main analyses. First, a basic assessment of the model performances is achieved with (1) Taylor diagrams, (2) Q-Q plots comparing the distributions of the observed and modelled current speed and

direction and (3) scatter diagrams uniquely for the AdriSC ROMS 1-km model, which is further used for the other analyses. Second, a climatological analysis of the AdriSC ROMS 1-km results is performed for the 7 different datasets (Fig. 1c) collected from experiments/scientific cruises described in Section 2.2.1. For each dataset, the following results are presented: (1) monthly climatology of the median (and MAD as upper and lower bounds) of the modelled and observed current speed and direction – this analysis is done without taking the same depth range for each month due to the vertical sparsity of the

measurements, (2) seasonal variations of the vertical profiles of the modelled and observed median current speed interpolated to standard oceanographic depths selected appropriately for each dataset – seasons are defined same as for the CTD analysis – and (3) seasonal variations of the modelled and observed current direction presented in the form of polar histograms (i.e. rose plots) showing the current direction distributions.

## 3 Results and Discussions

### 3.1 Modelled sea-surface properties

### 3.1.1 Evaluation

First, the main three normalized spatial EOF components (Fig. 2) and associated amplitude time series (Fig. 3), derived from the JPL_MEASURES SSHA gridded product and the AdriSC ROMS 3-km SSH results, are analysed and compared.

[Figure 2]

Overall, it can clearly be seen that, for both Adriatic and northern Ionian Seas (Fig. 2), the first EOF component (EOF1) represents the seasonal variability of both AdriSC ROMS 3-km and JPL_MEASURES results with spatial signal and amplitudes slightly stronger in the model (i.e. 81.2% of the total signal with amplitudes varying between ±8.0; Fig. 3) than in the observations (i.e. 74.5% of the total signal with amplitudes ranging between ±6.0; Fig. 3). The two remaining EOF





components are switched in the model compared to the observations (Figs. 2 and 3). In other words, the second ROMS 3-km

EOF component (EOF2 representing 6.2% of the total signal) corresponds to the third SSHA EOF component (EOF3 representing 3.0% of the signal) and vice versa. In addition, in the observations, the EOF2 component represents the decadal variability while the EOF3 component shows the interannual variability of the SSHA signal (Fig. 3). The comparison between modelled spatial EOF2 and observed spatial EOF3 (Fig. 2) reveals that, for the interannual variability, the AdriSC ROMS 3-km results don't reproduce the observed eddies in the northern Ionian Sea and present different spatial patterns in the north-

eastern Adriatic. Further, the interannual variability signal is generally stronger in the ROMS 3-km model (varying mostly between ±2.0; Fig. 3) than in the SSHA observations (ranging mostly between ±1.0; Fig. 3). Consequently, as both seasonal and interannual signals are stronger in the AdriSC ROMS 3-km results than in the JPL_MEASURES observations, the decadal variability and hence the so-called BiOS signal in the northern Ionian Sea (pattern clearly identified with strongly negative EOFs values; Figs. 2) is weaker in the model (2.1% of the total signal with amplitudes varying between ±1.5; Fig. 3) than in

the measurements (5.9% of the total signal with amplitudes ranging from ±2.0; Fig. 3). The differences between observation and modelling of the BiOS-driven signal can also be observed during the 2012-2014 period, after an intense dense water formation in 2012 (Mihanović et al., 2013) which had the capacity to reverse the circulation in the northern Ionian Sea (Gačić et al., 2014). Here, the modelled EOF3 reach a substantially negative values in 2012 and particularly in 2013, while the observations (EOF2) only present a slight decrease of amplitude which mostly stays positive during these two years. As such,

this may indicate a larger capacity of the dense waters to change the BiOS-driven patterns during these extremely severe years, compared to other BiOS-driven amplitude reversals (e.g. 1997, 2005 and 2009) which are of similar amplitude ratio in the ROMS 3-km model and the observations. Further, it may be hypothesized that the limited capability of the AdriSC ROMS 3-km model to reproduce the intensity of the BiOS signal is linked to the insufficient spatial extension of the ROMS 3-km domain to the south, where the boundary conditions thus have too much influence on the obtained results. Finally, despite

these limitations, it should also be noted that the time variations of the 3 main EOFs (Fig. 3) are overall well synchronized between the model and the observations.

[Figure 3]

Second, the AdriSC ROMS 3-km sea surface temperature results are compared to two different remote sensing products – i.e. AVHRR SST (Fig. 4) and JPL_MUR SST (Fig. 5) – with spatial maps of both the median of the gridded observations and the

median and 1st, 25th, 75th, 99th percentiles of the biases between the AdriSC ROMS 3-km results and the observations.

[Figure 4]

The spatial variations of the AVHRR median observations (Fig. 4) show that the lowest temperatures are present in the northern and western parts of the Adriatic Sea reaching around 17.0 °C on average. The middle- and south-eastern parts of the Adriatic have surface temperatures ranging from 17.5 to 19.0 °C. In the northern Ionian Sea median temperatures are higher, ranging





from 18.0 to 19.8 °C. Regarding the evaluation, the AdriSC ROMS 3-km model is underestimating the SST in the northern Adriatic, particularly along the plume of the Po river by down to -0.8 °C, while in the rest of the Adriatic biases are lower and ranging from -0.2 °C to 0.2 °C. In the Ionian Sea, the model tends to overestimate the SST by up to 0.6 °C along the western coast and by 0.1 °C on average along the eastern coast. In terms of the extreme underestimations, the biases reach down to - 2.0 °C in the northern Adriatic, -0.7 °C in the rest of the Adriatic and -0.5 °C in the Ionian Sea for the 25th percentile as well

as -4.0 °C in the northern Adriatic, -3.0 °C in the rest of the Adriatic and -2.0 °C in the Ionian Sea for the 1st percentile. Small negative biases down to -0.5 °C are still present in the northern Adriatic for the 75th percentile, with positive biases up to 0.3 °C for the rest of the Adriatic and up to 1 °C in the Ionian Sea. For the 99th percentile, the whole domain presents positive biases around 1.5 °C in the Adriatic and 2 °C in the Ionian Sea. It should be noted that, in some coastal parts of the domain, the observed dark blue patches are artefacts resulting from different representations of the coastline between the AdriSC ROMS

3-km model and the AVHRR remote sensing product.

[Figure 5]

The other SST dataset analysed in this study (JPL_MUR SST) has a shorter temporal span (i.e. only starts in June 2002) but a higher spatial resolution (i.e. 0.01° instead of 0.25°) and thus a more accurate representation of the coastline than AVHRR. The median of the JPL_MUR_SST dataset (Fig. 5) shows that the lowest temperatures are present in the northern and north-

eastern Adriatic reaching around 17.0 °C on average. The middle and western parts of the Adriatic have surface temperatures around 17.7 °C. The highest temperatures are in the middle- and south-eastern Adriatic ranging from 18.5 to 19.5 °C. In the northern Ionian Sea, median temperatures are mostly around 20.0 °C. Concerning the evaluation, the AdriSC ROMS 3-km model is generally underestimating the SST, except in the coastal north-eastern Adriatic and western part of the Ionian Sea. In the northern Adriatic, northern part of the western coast along the plume of the Po river and southernmost part of the eastern

coast, negative biases reach below -0.5 °C, while in the rest of the Adriatic as well as middle and eastern parts of the Ionian Sea biases reach down to -0.3 °C. A narrow strip of negative median biases may be seen along the eastern coast of the southern Adriatic, matching the plumes of the Albanian large rivers. In terms of the extreme conditions, for the 25th percentile, negative biases reach down to -2.0 °C in the northern Adriatic and northern part of the western coast, -1.0 °C in the rest of the Adriatic and -0.8 °C in the Ionian Sea. For the 1st percentile, biases reach down to -4.0 °C in the northern Adriatic, northern part of the

western coast and southernmost part of the eastern coast, -3.0 °C to -2.0 °C in the rest of the Adriatic and -3.0 °C in the Ionian Sea. For the 75th percentile, small negative biases are present in the northern Adriatic and northern part of the western coast down to -0.5 °C whereas for the rest of the Adriatic and Ionian Sea the temperature is overestimated by up to 0.3–0.8 °C. For the 99th percentile, the model overestimates the SST in the whole domain by up to 0.5–2.0 °C.





### 3.1.2 Discussion

This brief evaluation of the AdriSC ROMS 3-km sea-surface properties thus reveals that the model is capable to reproduce (1) the BiOS, even though with a weaker intensity due to the overestimation of both seasonal and interannual signals, and (2) the SST, despite presenting a persistent cold bias within the Adriatic Sea.

Within the Mediterranean climate community, the overall cold SST bias, present particularly during summer, is a well-known feature of the ocean models. First, following Akhtar et al. (2018) – which assessed the impact of model resolution and coupling
in the Mediterranean Sea – coupled atmosphere-ocean models are more likely to generate negative SST biases. Second, a comparison of SST results from six different models with remote sensing products (Darmaraki et al., 2019) has showed cold biases ranging from about −0.3 °C to −1.0 °C in average over the entire Mediterranean Sea. Finally, the cold summer SST biases are also known to be higher in the northern Adriatic Sea reaching below -3 °C on average (e.g. L'Hévéder et al., 2013; Di Luca et al., 2014; Sevault et al., 2014; Parras-Berrocal et al., 2020). Consequently, it can be safely said that the results
obtained with the AdriSC ROMS 3-km model are at least within the ranges of the known cold SST biases of the Mediterranean models. But, following these first results, the high-resolution AdriSC models also seem to improve the representation of the summer SST as the $25^{th}$ percentile – which most likely is representative of the summer month biases – only reaches a maximum value of -2 °C near the Po river and is -0.75 °C on average, over the entire Adriatic Sea.

As explained in Parras-Berrocal et al. (2020), the cold summer SST biases of the AdriSC ROMS 3-km model can result from
(1) a deficit of solar radiation by the AdriSC atmospheric model which have shown a systematic temperature underestimation (up to 5 °C) during the summer (Denamiel et al., 2021b), (2) some intrinsic shortcomings of the AdriSC ocean model such as vertical mixing, turbidity, etc. or (3) the fact that the river temperatures are imposed by taking the ERA-Interim skin temperatures the closest to the river estuaries, which is a crude approximation particularly for the largest Adriatic rivers such as the Po or the Albanian rivers. Finally, it is known that the optical properties of the water are playing a crucial role in
modelling the turbidity, which is responsible for most of the downward shortwave radiation absorption in the upper layer and thus potentially the presence of cold SST biases. The evaluation of the AdriSC ROMS 3-km SST results may thus also show the limitations of the implemented dQ/dSST procedure, which was supposed to mitigate the problems linked to the optical properties of the Adriatic waters.

### 3.2 Modelled thermohaline properties

### 3.2.1 Evaluation

The overall skills of AdriSC ROMS 3-km and 1-km models to reproduce the observed CTD data are presented in Figure 6.

[Figure 6]





First, correlations and normalized standardized deviations of modelled and observed temperature (Fig. 6a) and salinity (Fig. 6b) for each observational experiment and/or cruise are shown with Taylor diagrams. It should be noticed that the CSP01 experiment presents extremely small correlations for the temperature (0.0 and 0.3 for the AdriSC ROMS 3-km and 1-km models, respectively) and even anticorrelations for the salinity (-0.2 and -0.4 for the AdriSC ROMS 3-km and 1-km, respectively) which are for practical reasons marked as 0.0 on the Taylor diagram. Additionally, for the AdriSC ROMS 1-km model, the CSP01 experiment also presents large standardized deviations for both temperature and salinity (3.9 and 3.7, respectively) which are conveniently marked as 2.0 on the Taylor diagram. Since all the other datasets have relatively similar results, it is suspected that CSP01 may not be a reliable dataset for this evaluation and is treated as an outlier and removed from further analysis. For all the other observational experiments and/or cruises, the overall results (hereafter referred as All data) basically highlight that the AdriSC ROMS 3-km and 1-km models reproduced the observed temperatures with a good accuracy (i.e. correlations around 0.9 and normalized standardized deviations around 1.0) but do not properly capture the observed salinity (i.e. correlations around 0.7 and normalized standardized deviations between 0.3 and 0.5). This most probably highlights that even kilometre-scale ocean models struggle to accurately reproduce the fresh water input from the Adriatic rivers which are playing a crucial role in terms of the thermohaline circulation along the coasts (Vilibić et al., 2016, Vilibić et al., 2018). Second, the Q-Q plots of temperature and salinity (Fig. 6c and 6d) reveal that both models are capable to overall represent the observed distributions with only a small underestimation of the observed temperatures above 22 °C but a substantial overestimation of the observed salinity below 37.5. However, it should be noted that the number of records with salinity lower than 37.5 only represents less than 1 % of the entire dataset. Additionally, the AdriSC ROMS 1-km model presents significantly smaller salinity overestimations than the AdriSC ROMS 3-km model and is therefore solely used for further evaluation of the modelled thermohaline properties. Finally, the scatter plots (Fig. 6e and 6f) reveal that the hexagons with the largest number of points are following the reference line for both temperature and salinity, which indicates that the vast majority of the AdriSC ROMS 1-km results corresponds well to the observations in both intensity and timing.

[Figure 7]

A more detailed evaluation of the AdriSC ROMS 1-km thermohaline properties depending on four depth ranges (i.e. 0-50 m, 50-200 m, 200-500 m and 500-2000 m) is presented as the median (Fig. 7) and MAD (as supplementary material, Fig. S1) of the temperature and salinity biases (i.e. model minus observations), depending on the locations of the *in situ* observations. For the surface layer (0-50 m), the median temperature and salinity biases (Fig. 7a and 7b) present a large spatial variability. In general, a slight prevalence of temperature underestimation and salinity overestimation in the whole Adriatic can be noticed. Furthermore, biases are most pronounced in the northern Adriatic with dominant negative values for temperature, ranging from -4±0-0.9 °C to -2±0-0.9 °C, and dominant positive values for salinity, up to 4.3±0-1.4. The large overestimation of the salinity in the northern Adriatic upper layer is most probably influenced by the inaccurate representation of the river discharges in the model, especially of the Po River with the largest outflow in the Adriatic Sea (average discharge of 1500 m$^3$/s, Raicich, 1996;





Supić and Orlić, 1999). Strong negative temperature biases are also present in the middle Adriatic ranging from -2.4±0-0.4 °C
to -0.5±0-0.4 °C. Furthermore, larger median temperature and salinity biases are present in the north-eastern Adriatic coastal
areas, and particularly in the Kvarner Bay, ranging from -2.0±0.0-1.2 °C to 1.3±0.0-1.2 °C and -0.7±0.0-1.0 to 2.9±0.0-1.0,
respectively. Biases in the western coastal part of the Adriatic range from -1.0±0.0-0.3 °C to 1.5±0.0-0.3 °C for the temperature
and -0.8±0.0-0.2 to 1.5±0.0-0.2 for the salinity. In the middle eastern (including the Dalmatian Islands) and the southern

Adriatic, biases are of the order of -1.8±0.0-0.9 °C to 0.8±0.0-0.9 °C for the temperature and -0.9±0.0-1.0 to 1.7±0.0-1.0 (only
-0.4±0.0 to 0.3±0.0 in the southern Adriatic) for the salinity.

For the upper intermediate layer (50-200 m), the model reproduces dominantly warmer temperatures, except in the southern
Adriatic where the biases range between -0.7±0.0-0.2 °C and 0.4±0.0-0.2 °C (Fig. 7c). The largest temperature overestimations
are located in the north-eastern Adriatic with biases up to 3.0±0.8 °C, as well as in the middle Adriatic and the middle eastern

coastal areas with biases up to 1.5±0.8 °C. Concerning the salinity, the model generally underestimates the observations (Fig.
7d) except in the north-eastern Adriatic where positive salinity biases up to 0.5±0-0.1 are dominant.

For the lower intermediate layer (200-500 m), positive temperature biases up to 1.5±0.0-0.2 °C adjacent to negative biases
down to -1.0±0.0-0.2 °C are located in the middle Adriatic and more precisely in the Jabuka Pit, the collector of the northern
Adriatic dense waters (Vilibić and Supić, 2005), while negative biases down to -0.8±0.01-0.2 °C are present in the southern

Adriatic (Fig. 7e). Salinity biases are slightly negative with values down to -0.1±0.0 (Fig. 7f). For the deeper layers (500-2000
m), temperature is underestimated in the southern Adriatic Pit as well as in the northern Ionian Sea with biases ranging from -
0.1±0.1 °C to -0.8±0.1 °C (Fig. 7g). Similarly, salinity biases are relatively low and range from -0.1±0.0 to 0.0±0.0 (Fig.7h).
Overall, the spatial analysis of the CTD stations depending on the depth reveals that the capability of AdriSC ROMS 1-km
model in reproducing temperature and salinity is generally better in deeper parts of the Adriatic than in the coastal areas and

the shallow northern Adriatic shelf.

The last kind of analysis is an in-depth climatological and seasonal evaluation of the AdriSC ROMS 1-km thermohaline
properties performed for seven predefined subdomains (Fig. 1b). However, to keep a reasonable article length, only three of
these subdomains are fully analysed hereafter (Figs. 8 to 10) while, for the remaining four subdomains, only a summary is
presented and the full description is provided as supplementary material (Figs. S2 to S5).

455                                                                    [Figure 8]

For the Northern Adriatic subdomain (Fig. 8), the AdriSC ROMS 1-km model is overall lacking of accuracy in reproducing
the thermohaline properties. The monthly temperature climatology is reproduced relatively well most of the year (i.e. biases
ranging from -0.6±1.8 °C to 0.6±0.3 °C) except in August, September and October when the differences can reach down to -
1.6±1.3 °C (Fig. 8a). The salinity is not well represented, especially in the second half of the year with persistently higher



values by up to 0.7±0.3 (Fig. 8b). Regarding the seasonal variations, the vertical profiles of the median temperature biases
show a strong underestimation reaching down to -1.0 °C in spring and -2.0 °C in summer in the surface layer (Fig. 8d).
However, during these two seasons, a large overestimation of the temperature is present below 10 m and up to 3.8 °C at 30 m
depth. In autumn, temperature is mostly negatively biased and the underestimation reaches down to -0.7 °C. Winter
temperature biases are rather small throughout the whole water column. In addition, salinity is strongly overestimated at the

surface independently of the season (Fig. 8e) with winter having the smallest biases (below 0.5) and summer having the largest
biases (up to 2.1). Below 20 m the salinity biases are smaller and the seasonal variability is weaker. The analysis of the T-S
diagrams reveals that the model performs well in reproducing dense water masses and since the northern Adriatic is a well-
known and one of the most researched dense water formation sites (Zore-Armanda, 1963; Vilibić and Supić, 2005; Mihanović
et al., 2013, 2018, Vilibić et al., 2016), these results are promising. The model seems to be less accurate in the density ranges

below 25 kg m$^{-3}$ in which there is an overestimation of density (Fig. 8g, 8h). In addition, despite the lack of accuracy of the
model for salinities under 36 and temperatures over 24 °C, most of the observations are well represented in the T-S diagram
with the ROMS 1-km model.

For the Western Coast subdomain (Fig. 9), the AdriSC ROMS 1-km model seems to well represent the monthly temperature
climatology (Fig. 9a) despite a tendency to higher positive biases from May to November (up to 1.0±0.8°C to 2.0±0.2°C). To

be noted, the highest bias is found in July when the amount of available data is quite small (Fig. 9c). Furthermore, salinity
climatology is reproduced with a good accuracy throughout the whole year (Fig. 9b). Seasonally, the strongest temperature
biases are observed in summer and autumn: (1) mostly negative in the first 20 m where they reach down to -1.0 °C, and (2)
becoming positive below 20 m (up to 1.0 °C) till 200 m where they decrease (Fig. 9d). Winter temperature biases are generally
small and decrease with the depth to reach nearly 0 °C below 100 m. In spring, the temperature biases are negative in the

surface but becomes positive below 20 m down to 100 m similarly to the other seasons. Salinity biases seem to be the strongest
in the surface with an underestimation of -1.0 in spring and an overestimation up to 1.5 in winter (Fig. 9e). However, a very
small number of observations were recorded at this depth (Fig. 9f). Additionally, salinity biases are small throughout the water
column independently of the season. Finally, the seasonal analysis of the T-S diagrams shows that the AdriSC ROMS 1-km
model is capable to reproduce the seasonal properties of the Western Coast subdomain water masses (Fig. 9g, 9h), where the

outflow of freshened waters from the northern Adriatic is occurring (Artegiani et al., 1997; Lipizer et al., 2014; Burrage et al.,
2009).

[Figure 10]

For the Deep Adriatic subdomain (Fig. 10), the modelled monthly temperature and salinity medians are lower than the
observations throughout the whole year. The highest biases occur in winter and spring reaching almost -0.7±0.2°C for the

temperature (Fig. 10a), while the differences in salinity are smaller than -0.1±0.0 (Fig. 10b). Seasonal analysis shows that the
temperature biases are mostly negative, down to -2.0 °C in the surface in summer and associated with a small number of





observations (Fig. 10d) while below 50 m they are mostly smaller than -0.5 °C (Fig. 10f). More precisely, the underestimation of the observations is minimized between 50 m and 300 m of depth for all seasons, except in winter when the biases reach down to -0.5 °C. However, stronger temperature underestimations are present in the deeper layers between 300 m and 900 m

of depth but rapidly decrease below 900 m of depth. Salinity is overestimated in summer and winter in the surface layer and mostly underestimated for all the other depths and seasons with biases smaller than -0.1 (Fig. 10e). The seasonal analysis of the T-S diagrams shows that the model performs well independently of the season, with slightly narrower temperature and salinity ranges and an overestimation of densities under 26.5 kg m$^{-3}$, particularly in summer (Fig. 10g, 10h). The densest waters are captured relatively well which is important as the Deep Adriatic subdomain is a well-known dense water formation site

(Vilibić and Orlić, 2001, 2002; Manca et al., 2002; Mantziafou and Lascaratos, 2004, 2008).

For the other subdomains (i.e. Middle Adriatic, Otranto-Ionian, Kvarner Bay and Dalmatian Islands; Figs. S2 to S5) a detailed analysis is presented as supplementary material but can be briefly summarized as follows. Monthly climatologies of temperature and salinity are well captured in the Middle Adriatic and Otranto-Ionian, while the latter subdomain has slightly negative biases. In the Kvarner Bay and Dalmatian Islands subdomains, the AdriSC ROMS 1-km model is capable to reproduce

the temperature monthly climatology for the entire year except summer. Salinity is captured relatively well in the Kvarner Bay with certain overestimation, whereas in the Dalmatian Islands subdomain it is slightly underestimated. Vertical profiles of temperature and salinity in the Middle Adriatic subdomain reveal that the temperature biases are mostly negative in autumn and positive in winter, while the salinity biases are generally negative except in summer at 10 m depth. In the Kvarner Bay subdomain, vertical profiles of temperature are best reproduced in autumn when the biases are very small, while for other

seasons there is an overestimation. However, salinity is overestimated the entire year in the whole water column with higher biases in the surface layer. The Dalmatian Islands subdomain has the largest positive temperature biases in summer and smallest biases in winter and spring, while the largest salinity underestimations occur in summer and autumn. Lastly, for the Otranto-Ionian subdomain, largest variations of temperature biases are present down to 200 m. Below this depth the biases are similar and negative for all the seasons. Salinity biases are largest between 100 and 200 m, while below this layer the biases

are very small. Concerning the T-S diagrams, the model performs well for all the subdomains independently of the season with a common overestimation of the densities lower than 26 kg m$^{-3}$.

### 3.2.2 Discussion

In summary, the evaluation of the AdriSC ROMS 1-km thermohaline properties shows that the model is overall capable to reproduce the temperature and salinity in all of the analysed subdomains and mostly with a good accuracy. In the middle

Adriatic, the western coast and the middle-eastern coastal parts of the Adriatic (e.g. Dalmatian Islands sub-domain), monthly climatologies are well represented, whereas the largest biases are found in the surface layer (up to 50 m of depth) during summer with maximum ±1.0 °C for the temperature and ±0.2 for the salinity. These are most probably linked to the quoted problems with the optical properties of the Adriatic waters and/or the river discharges. Additionally, in the deepest parts of the





southern Adriatic as well as the Strait of Otranto and the northernmost part of the Ionian Sea, biases are persistently negative
in the temperature, by about -0.25 °C on average, while the salinity biases are lower than ±0.1.

However, in some areas at certain depths and depending on the time of the year, the AdriSC ROMS 1-km model lacks of
accuracy. In general, the largest differences are found in the northern Adriatic, with negative temperature biases in summer
(down to -2.0 °C) associated with large positive salinity biases (up to 2.0) in the surface layer. As seen previously for the SST
evaluation, the cold bias is probably linked to the improper estimation of the Po river temperature while the overestimation of
the salinity for the lowest values probably comes from the improper estimation of the Po freshwater fluxes. As similar results
(i.e. overestimation of surface salinity and overestimation of the summer temperature in surface) are also found in the north-
eastern coastal part of the Adriatic and a large scatter of the lowest salinity values is present in Figure 6, the AdriSC ROMS
models seem to struggle to reproduce the proper river plume dynamics in the northern Adriatic. Nevertheless, these results are
still outperforming the ones of the previous Mediterranean RCMs evaluated in the Adriatic Sea, which exhibited biases above
3.0 for the salinity and below -3.0 °C for the temperature in the northern Adriatic (e.g. L'Hévéder et al., 2013; Di Luca et al.,
2014; Sevault et al., 2014; Parras-Berrocal et al., 2020). Additionally, independently of the subdomains, the analysis of the
vertical profiles shows that the temperature and salinity biases often present a peak in the vicinity of the thermocline depth
which can probably be linked to an inaccurate representation of vertical diffusivity and vertical mixing in the AdriSC ROMS
models. Finally, at the Jabuka Pit, where strong positive temperature biases are adjacent to negative ones, the representation
of the bathymetry by the model (e.g. 1-km resolution, flattening due to the smoothing procedure) may have impacted the
location and the amount of dense water collected. However, it should be noted that within the Middle Adriatic subdomain,
which includes the Jabuka Pit, the coldest more saline waters are well represented by the AdriSC ROMS 1-km model as seen
in the T-S diagram.

### 3.3 Modelled dynamical properties

**3.3.1 Evaluation**

A basic skill assessement of the AdriSC ROMS 3-km and 1-km models to reproduce the observed ADCP and RCM hourly
measurements is presented in Figure 11.

[Figure 11]

First, correlations and normalized standardized deviations of the modelled and observed ocean current speeds (Fig.11a) and
directions (Fig.11b) for each dataset are shown with Taylor diagrams. Following these analyses, the AdriSC ROMS 3-km and
1-km models seem to reproduce the observed correlations between 0.2–0.5 and normalized standardized deviations ranging
from 0.5–1.7 for the current speeds as well as correlations around 0.2 and normalized standardized deviations between 0.7–
1.1 for the current directions. However, the Q-Q plot analyses of current speeds and directions (Fig. 11c and 11d) reveal that



both models are in fact perfectly capable to represent the observed distributions, except for a small overestimation of the

current speeds above 0.5 m s⁻¹. Consequently, the low correlations obtained for the Taylor diagrams must have been uniquely

linked to a lack of synchronization between hourly observations and model results. It should also be noted that the number of

records with speeds higher than 0.5 m s⁻¹ represents less than 1% of the entire dataset. Additionally, the current speed

overestimations are smaller for the AdriSC ROMS 1-km results than for those of the AdriSC ROMS 3-km model. Therefore,

the AdriSC 1-km model is solely used for further evaluation of the modelled dynamical parameters. Finally, the scatter plot

analyses (Fig. 11e and 11f) show that the hexagons with the highest density of records are overall following the reference line

for both current speeds and directions. However, due to the already mentioned lack of synchronization, modelled current

speeds and most especially modelled current directions can be extremely spread compared to the observations. Despite the

inherent difficulties to reproduce the ocean dynamics at the hourly scale, the scattering of the AdriSC ROMS 1-km results can

also result from the uncertainties linked to the observational dataset time references. Indeed, due to the lack of metadata

availability for a certain number of datasets, some observations which may have been provided in local time have been

compared with model results in Universal Time Coordinated (UTC). Further, it should be noted that the two vertical lines

produced on the current direction scatter plot are in fact inconsistencies identified in the JP2 dataset for 2 stations and are

removed from further analyses.

The last kind of analyses is an in-depth climatological and seasonal evaluation of the AdriSC ROMS 1-km dynamical

properties performed for seven different datasets (Fig. 1c). However, to keep a reasonable article length, only three of these

datasets are fully analysed hereafter (Figs. 12 to 14) while, for the remaining four datasets, only a summary is presented and

the full description is provided as supplementary material (Figs. S7 to S10).

[Figure 12]

For the DART_ADCP dataset (Fig. 12), the monthly climatology differences of the AdriSC ROMS 1-km and the observed

current speeds can reach up to 0.02±0.01 m s⁻¹ in October and down to -0.05±0.02 m s⁻¹ in March (Fig. 12a) while the direction

differences reach down to -39±48 ° in September. Concerning the seasonal variations, the vertical profiles of the modelled and

observed current speeds (Fig. 12d) show an underestimation reaching -0.03 m s⁻¹ in winter and -0.02 m s⁻¹ in spring. The

highest differences occur in autumn reaching down to -0.05 m s⁻¹, while in summer very low biases are present throughout the

water column except at 5 m depth. The rose plots (Fig. 12e) of the modelled and observed current direction reveal that the

observed distributions are similarly reproduced by the model independently of the season. It can be seen that the occurrences

of the eastward current direction are slightly overestimated, while the north-eastward direction is underestimated.

[Figure 13]



For the JP2 dataset (Fig. 13), AdriSC ROMS 1-km reproduced the monthly climatology of current speed with good accuracy (Fig. 13a) which is supported with a large number of observations throughout the year (Fig. 13c). The largest difference of 0.02±0.01 m s⁻¹ is reached in November. However, the current direction climatology is mostly overestimated reaching up to 87.43±7.16 ° in June (Fig. 13b). Seasonal vertical profiles of the modelled and observed current speed (Fig. 13d) show an overestimation under 10 m in winter and autumn reaching up to 0.03 m s⁻¹. Extremely low biases are present in spring and summer down to 40 m depth below, where they reach 0.02 m s⁻¹. According to the rose plots (Fig. 13e), the main current directions within this dataset are well reproduced for all seasons and with a slight overestimation of occurrences of all directions, except of the eastward direction which are strongly underestimated independently of the season. This systematic underestimation may be ascribed to the inaccurate representation of the coastline in the model at the locations of the extracted points.

[Figure 14]

For the IOR_Data_RCM dataset (Fig. 14), the monthly climatology of current speed in summer and autumn is well represented by AdriSC ROMS 1-km, while the differences are more pronounced in winter and spring varying between -0.03±0.02 m s⁻¹ and 0.03±0.02 m s⁻¹ (Fig. 14a). The current direction climatology is reproduced with good accuracy by the model, except in winter when the biases reach up to 50±29 ° in January (Fig. 14b). Regarding the seasonal variations, the vertical profiles of the modelled speed are generally in good agreement with the observed speed in the first 40 m (Fig. 14d). Within this layer, very small differences are present in winter and autumn, while in summer and spring the model tends to underestimate the observed speed down to -0.02 m s⁻¹. Below this depth, the observations are underestimated for all seasons. To be noted, the number of observations is largest in the first 50 m, whereas 99.5 % of all the data is concentrated within the first 100 m. The rest of the data (i.e. 0.5 %) is spread between 100 m and 900 m of depth, thus only the first 100 m are presented on the vertical plots. Lastly, the rose plots of the modelled and observed current direction (Fig. 14e) reveal that the observed distributions are similarly reproduced by the model independently of the season. The direction differences are slightly larger in autumn for the eastern currents.

For the other datasets (Figs. S6 to S9), monthly climatology as well as the seasonal vertical profiles of the current speed are mostly underestimated by the AdriSC ROMS 1-km model for the JP1 and IOR_Pal_ADCP dataset. Monthly climatology of the current direction for the JP1 dataset has small positive differences and the distributions show an overestimation of the south-eastward and north-westward current directions in summer and autumn. For the IOR_Pal_ADCP dataset, current direction monthly climatology is mostly underestimated independently of the season, while the main eastward and south-eastward current directions are overestimated. The current speed climatology and vertical profiles are generally well reproduced for the NAdEx_ADCP dataset, as well as the current direction with small differences between the model and observations. Finally, for the EACE dataset, current speed monthly climatology and vertical profiles show an overestimation in winter and underestimation in summer. The current direction monthly climatology is largely underestimated in winter,





whereas the biases are small in spring and autumn. According to the distributions, this is particularly true for the main north-westward current direction.

### 3.3.2 Discussion

In summary, the evaluation of the AdriSC ROMS 1-km dynamical properties reveals that the model is overall in good agreement with the observed hourly ocean current speed and direction. In general, there is a certain mismatch in time of the

model results and the observations which may be ascribed to a lack of synchronization between hourly observations and model results as well as to the uncertainties linked to the observational dataset time references. This demonstrates the inherent difficulties to reproduce the ocean dynamics and to evaluate the model results at the hourly scale.

Concerning the datasets, the RCM measurements (i.e. IOR_Data_RCM dataset), which are located mostly along the eastern coast (including islands) and at some offshore locations, are well reproduced by the model with relatively small biases (up to

±0.03 m s$^{-1}$ for the speed and a maximum of 50 ° for the direction). The ADCP measurements of current speed in the middle-eastern coastal area (i.e. JP1 and JP2 datasets) are relatively well captured (biases up to 0.04 m s$^{-1}$), while more significant differences are obtained for the current direction (biases up to 87 °). Additionally, a systematic underestimation of the occurences of the main current directions may be linked to a misrepresentation of the coastline in the model at certain locations. Furthermore, ocean current measurements along the transect across the Palagruža Sill (i.e. DART_ADCP and IOR_Pal_ADCP

datasets) are modelled with a general underestimation of current speed (down to -0.05 m s$^{-1}$) and an overestimation of the occurrences of the main current direction, which can be linked to the bathymetry representation in the model (e.g. 1-km resolution, smoothing procedure, etc.). Lastly, the ADCP measurements in the north-eastern part of the Adriatic (i.e. NAdEx_ADCP dataset) are reproduced mostly with a good accuracy, but with a slight underestimation of the current speed (down to -0.02 m s$^{-1}$). However, a significant improvement is achieved compared to the results of the ALADIN/ROMS

modelling system which was evaluated on the same set of measurements from the NAdEx experiment (Vilibić et al., 2018). Indeed, the authors have shown that the model strongly underestimated the observed current speeds by 50-80 % on average while AdriSC ROMS 1-km underestimates current speed by only 18 % on average. This highlights that, in the north-eastern Adriatic, higher horizontal and vertical ocean and atmospheric model resolutions, better resolving the complex bathymetry and orography, are required to reproduce the mesoscale variability of the winds and particlularly the hurricane strength bora

winds.

### 4 Summary and perspectives

In the presented study, the evaluation of the AdriSC ROMS 3-km and 1-km ocean models – forced by the already evaluated AdriSC WRF 3-km model (Denamiel et al., 2021b) – has been carried out for the 31-year long period (1987-2017). The main novelties of the work are, first, the implementation for the very first time – at least to the author's knowledge – of a kilometre-



scale coupled atmosphere-ocean model for long-term climate studies and, second, the amount of *in situ* data collected to
perform the evaluation of both daily thermohaline (CTD measurements) and hourly dynamical (ADCP and RCM observations)
properties of the AdriSC ocean models.

The findings of the evaluation are fourfold. First, the AdriSC ROMS 3-km model has been found to show a skill in reproducing
(1) the observed decadal signal of sea-surface height anomaly interpreted as the BiOS cycles – despite presenting a weaker
intensity compared to the seasonal and interannual variabilities, and (2) the observed SST – despite presenting a persistent
negative bias within the Adriatic Sea probably linked with the summer cold bias found in the AdriSC WRF 3-km model
(Denamiel et al., 2021b). Second, the AdriSC ROMS 1-km model has been found to be more suitable to reproduce the observed
daily temperatures and salinities as well as hourly ocean currents than the AdriSC ROMS 3-km model, thus highlighting the
necessity for higher resolution ocean climate simulations in the Adriatic Sea. Then, the detailed analysis of the AdriSC ROMS
1-km simulation revealed that (1) for the daily temperature and salinity, better results are found in the deepest parts than in the
shallow shelf and coastal parts, particularly for the surface layer of the Adriatic Sea, while, (2) for the hourly ocean currents,
better results are found for the RCMs and ADCPs located along the eastern coast and the north-eastern shelf than for the
ADCPs located in the middle-eastern coastal area and the deepest part of the Adriatic Sea. Finally, the AdriSC ROMS 1-km
model was found (1) to perform well in reproducing the seasonal thermohaline properties of the water masses over the entire
Adriatic Sea, despite a common overestimation of PDAs lower than 26 kg m$^{-3}$, and (2) consequently, to be a suitable modelling
framework for studying the long-term thermohaline circulation triggered by the dense waters forming in the northern Adriatic
Sea, cascading along the Italian coast and reaching the northern Ionian Sea where they potentially influence the BiOS regimes.
Additionally, it can also be envisioned to study these processes for a far future period (i.e. 2070-2100 period) with the AdriSC
long-term projections under climate change scenarios following the Pseudo-Global Warming (PGW; Schär et al., 1996)
method. This method has already been tested successfully with the AdriSC model for an ensemble of short-term extreme events
in the Adriatic Sea (Denamiel et al., 2020a, 2020b). Therefore, the AdriSC climate simulations are expected to broaden the
knowledge about the dynamics of the Adriatic-Ionian region.

Another important issue raised by this study is that a proper comparison of the ocean climate model skills in the Mediterranean
is particularly difficult to achieve due to the absence of standardized ocean observational datasets (similar to the E-OBS
products in the atmosphere; https://surfobs.climate.copernicus.eu/dataaccess/access_eobs.php). Instead, ocean models are
evaluated at different spatial and temporal ranges based on the observational datasets available to given researchers of given
countries, which makes a fair comparison between models almost impossible. Therefore, inter-comparing ocean climate
models in the Mediterranean could only be achieved through the creation of such standardized datasets and, consequently, a
change of the ocean data sharing policies, at least at the European level.

Finally, finding the right balance between numerical model accuracy (i.e. resolution) and efficiency (i.e. computational
resources and running time) – depending of the temporal and spatial scales of the studied processes – remains one of the major

issues of the climate modelling community. For example, (1) the RCMs of the Med-CORDEX community have already been proven to largely underestimate the dense water budget of the Adriatic Sea (Dunić et al., 2019), while (2) the recently developed MEDSEA ocean re-analysis at approximately 4-5 km (Escudier et al., 2020) is forced by the ERA5 atmospheric re-analysis
known to underestimate the extreme bora events (Denamiel et al., 2021a). To properly capture the Adriatic thermohaline circulation triggered by the dense water formation in the northern Adriatic Sea, the reliability of MEDSEA in the Adriatic Sea thus largely depends on the data assimilation and not the physics. Additionally, even the 1-km resolution used for the AdriSC ocean model is still quite coarse to study the consequences of extreme events along the Adriatic coasts such as flooding in Venice (Denamiel et al., 2020a) or meteotsunamis in Vela Luka (Denamiel et al., 2019). Consequently, it can even be
envisioned to downscale the AdriSC climate results during extreme events to a 1.5-km resolution in the atmosphere and up to 10 m (with an unstructured grid) along the coastal areas in the ocean, following the setup of some operational models in the Adriatic (e.g. for coastal floods, Umgiesser et al., 2020).

In conclusion, the added value of high-resolution coastal models in climate research of complex coastal regions such as the Adriatic, has been evidently demonstrated in this study. The main challenges which include high computational cost and
slowness of the models, are still actual but may be overcome in a near future due to the constant technological and scientific advancements.

**Code availability**

The code of the COAWST model as well as the ecFlow pre-processing scripts and the input data needed to re-run the AdriSC climate model in evaluation mode for the 1987-2017 period can be obtained under the Open Science Framework (OSF) FAIR
data repository https://osf.io/zb3cm/ (doi:10.17605/OSF.IO/ZB3CM).

**Data availability**

The model results and the measurements as well as the post-processing scripts used to produce this article can be obtained under the Open Science Framework (OSF) FAIR data repository https://osf.io/w8f4j/ (doi: 10.17605/OSF.IO/W8F4J).

**Author contribution**

IV and CD defined concept and design of the study. Set-up of the model and simulations were performed by CD. Material preparation was done by CD and PP. Production of the figures was done by PP and CD. Analysis of the results was performed by CD, PP and IV. The first draft of the manuscript was written by PP. All authors were engaged in commenting, revising and polishing of the manuscript. All authors read and approved the final manuscript.



**Competing interests**

The authors declare that they have no conflict of interest.

**Acknowledgments**

The contribution of all the organisations that provided the observations used in this study – the Istituto Nazionale di Oceanografia e di Geofisica Sperimentale (OGS), the Institute of Oceanography and Fisheries (IOR), the Ifremer (http://www.ifremer.fr/medar/cdrom_database.htm), the SACLANT Undersea Research Centre, the Consiglio Nazionale delle

Ricerche (CNR), Istituto di Biologia del Mare Venice, the Hellenic Centre for Marine Research, the Hellenic National Oceanographic Data Centre (HCMR/HNODC) and the Ruđer Bošković Institute – is acknowledged. Acknowledgement is also made for the support of the European Centre for Middle-range Weather Forecast (ECMWF) staff as well as for ECMWF's computing and archive facilities used in this research. This work has been supported by projects ADIOS (Croatian Science Foundation Grant IP-2016-06-1955), BivACME (Croatian Science Foundation Grant IP-2019-04-8542), CHANGE WE

CARE (Interreg Croatia-Italy program) and ECMWF Special Projects (The Adriatic decadal and interannual oscillations: modelling component, and Numerical modelling of the Adriatic-Ionian decadal and interannual oscillations: from realistic simulations to process-oriented experiments).

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

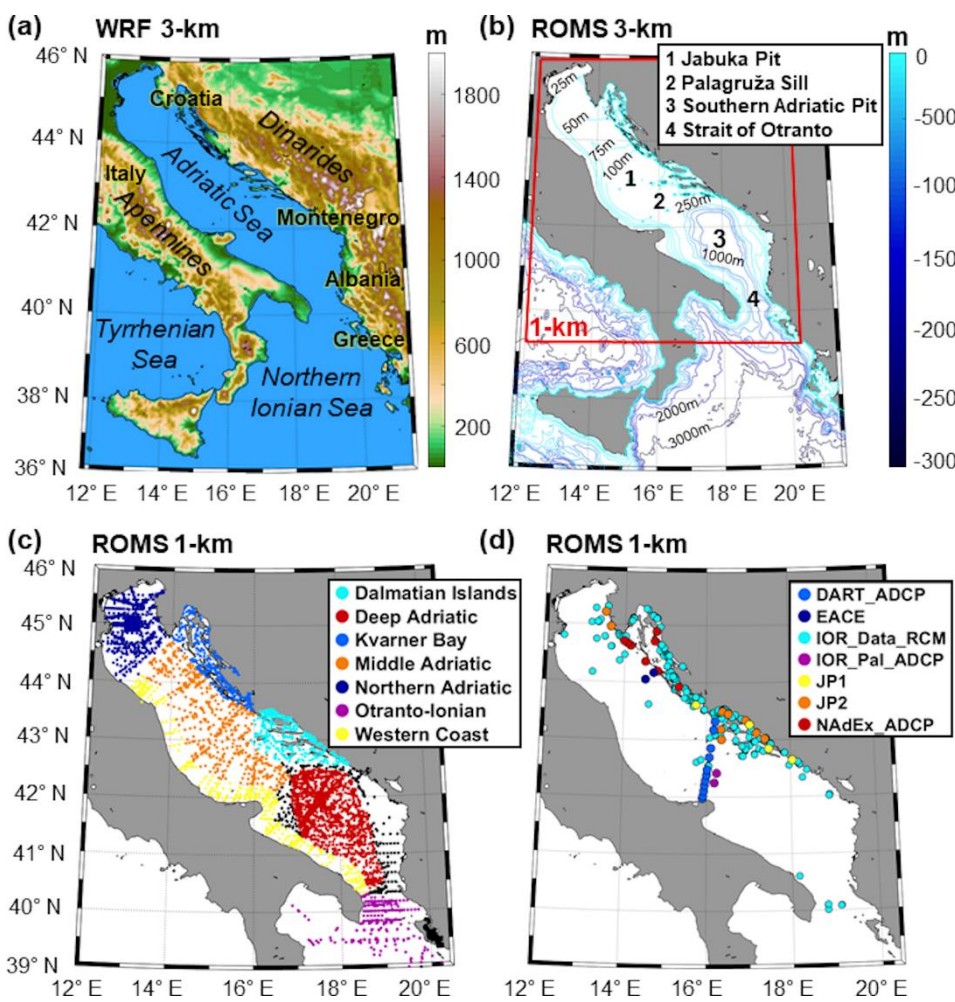

**Figure 1. (a)** AdriSC WRF 3-km domain and orography with geographical locations **(b)** AdriSC ROMS 3-km and ROMS 1-km domains and bathymetry as well as location of both **(c)** Conductivity Temperature Depth (CTD) observations separated in 7 sub-domains and **(d)** Acoustic Doppler Current Profiler (ADCP) or Rotor Current Meter (RCM) measurements from 7 different sources.





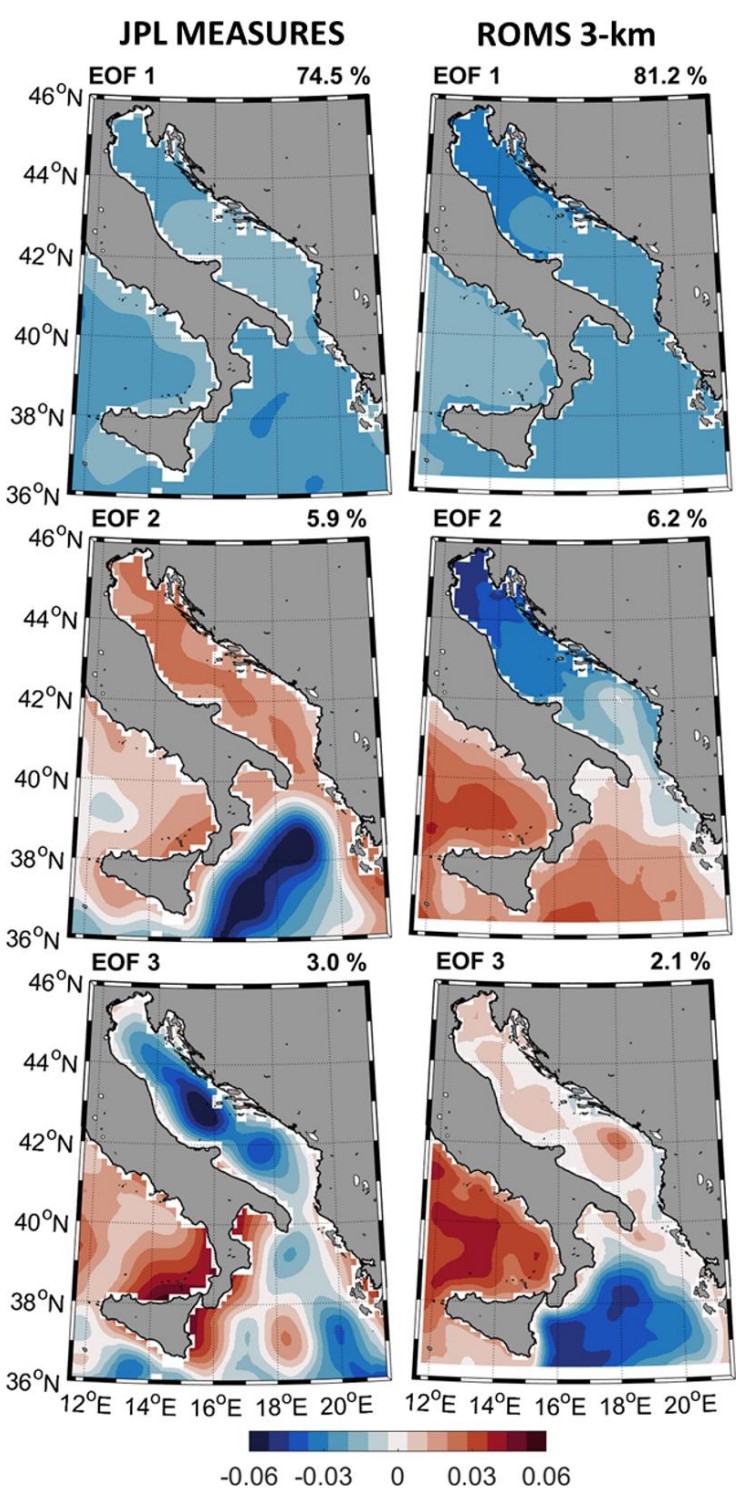

**Figure 2. Main three normalized spatial EOF components derived during the 1993-2017 period from the SSHA from the JPL MEASURES gridded product (left panels) and the SSH results of the AdriSC ROMS 3-km model (right panels).**






**Figure 3. Time series of the amplitudes associated with the main three normalized spatial EOF components derived during the 1993-2017 period from the SSHA from the JPL MEASURES gridded product (left panels) and the SSH results of the AdriSC ROMS 3-km model (right panels).**





**Figure 4. Median of the AVHRR daily sea-surface temperature product (top left panel) as well as median (top right panel) and 25th (centre left panel), 75th (centre right panel), 1st (bottom left panel), 99th (bottom right panel) percentiles of the daily sea-surface temperature biases between AdriSC ROMS 3-km model results and AVHRR product during the 1987-2017 period.**





**Figure 5.** As in Fig. 4 but for the JPL MUR daily sea-surface temperature product during the 2002-2017 period.






**Figure 6.** Evaluation of the AdriSC ROMS 3-km and 1-km temperature (left panels) and salinity (right panels) results against observations from 17 different datasets with (a, b) Taylor diagrams and (c, d) quantile–quantile plots as well as, only for the 1-km model, (e, f) scatter plots showing the density (number of occurrences) with hexagonal bins and total number of points n.

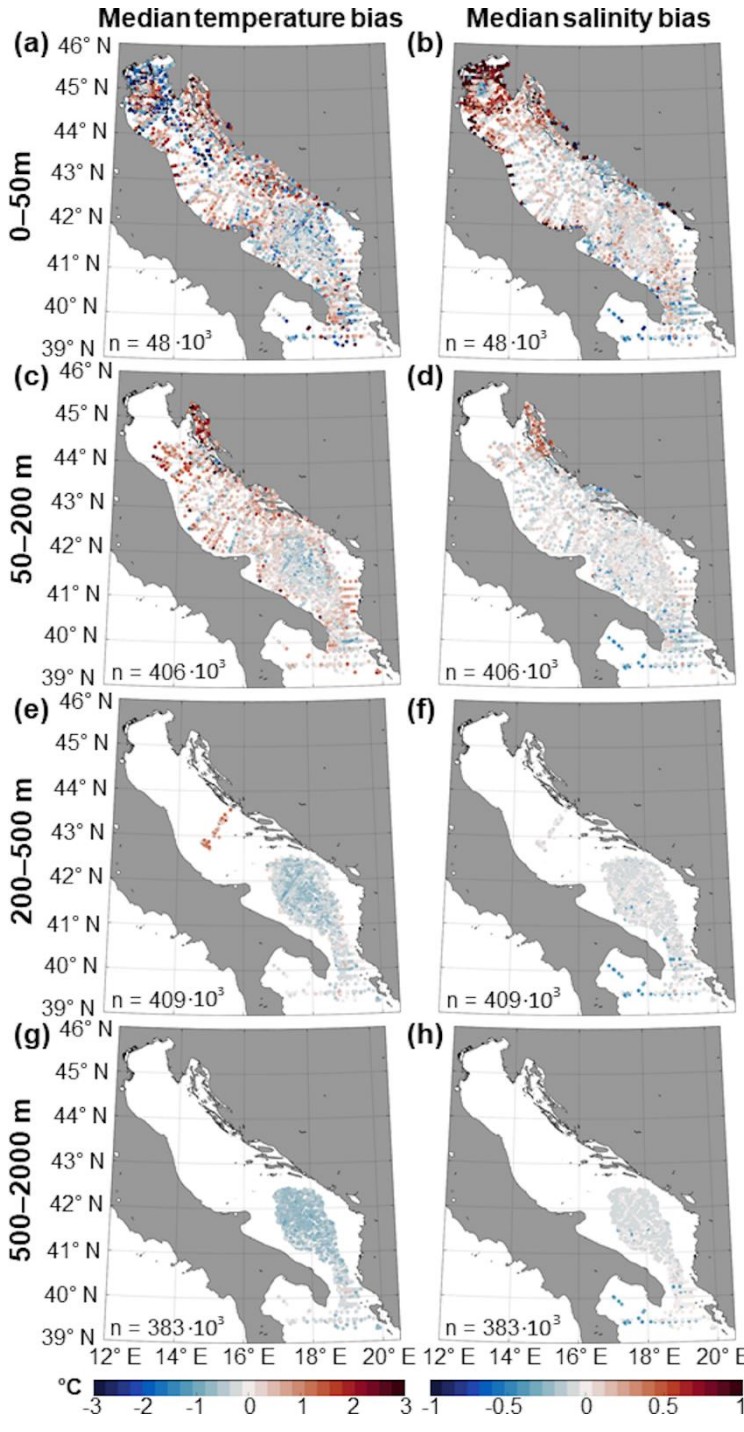


**Figure 7. Median of the temperature (left panels) and salinity (right panels) biases between AdriSC ROMS 1-km model results and CTD observations for depth ranges: (a, b) 0-50 m, (c, d) 50-200 m, (e, f) 200-500 m, (g, h) 500-2000 m, with the total number of points n (bottom left corner).**



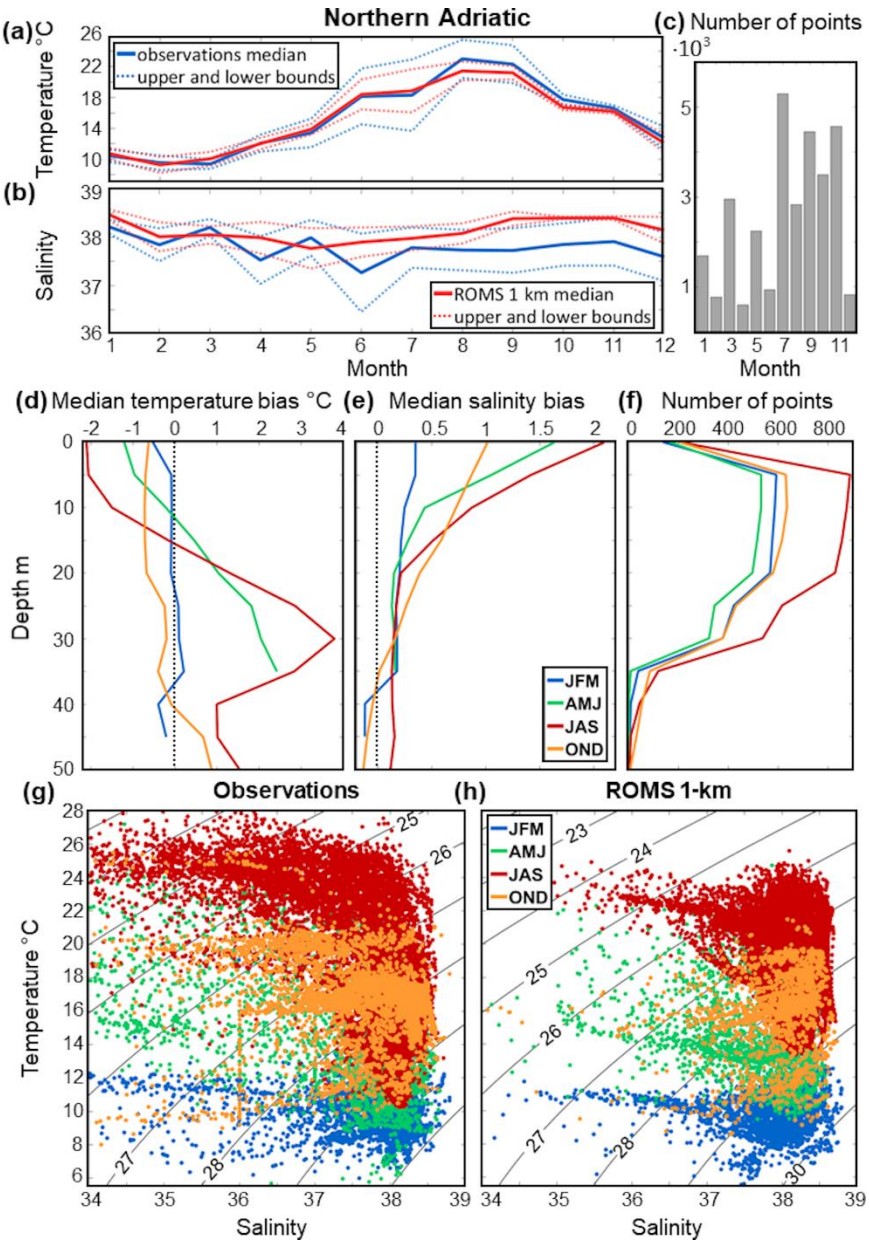

**Figure 8. Northern Adriatic subdomain.** Monthly climatology of AdriSC 1-km and *in situ* **(a)** median temperature, **(b)** median salinity and their variabilities (i.e. upper and lower bounds defined as ±MAD) as well as **(c)** number of observations per month. Seasonal variations of the **(d)** temperature and **(e)** salinity biases between the AdriSC ROMS 1-km model and observations depending on the depth as well as **(f)** number of observations per depth. Seasonal T-S diagrams for **(g)** the CTD observations and **(h)** the AdriSC ROMS 1-km model with Potential Density Anomaly (PDA) isolines.





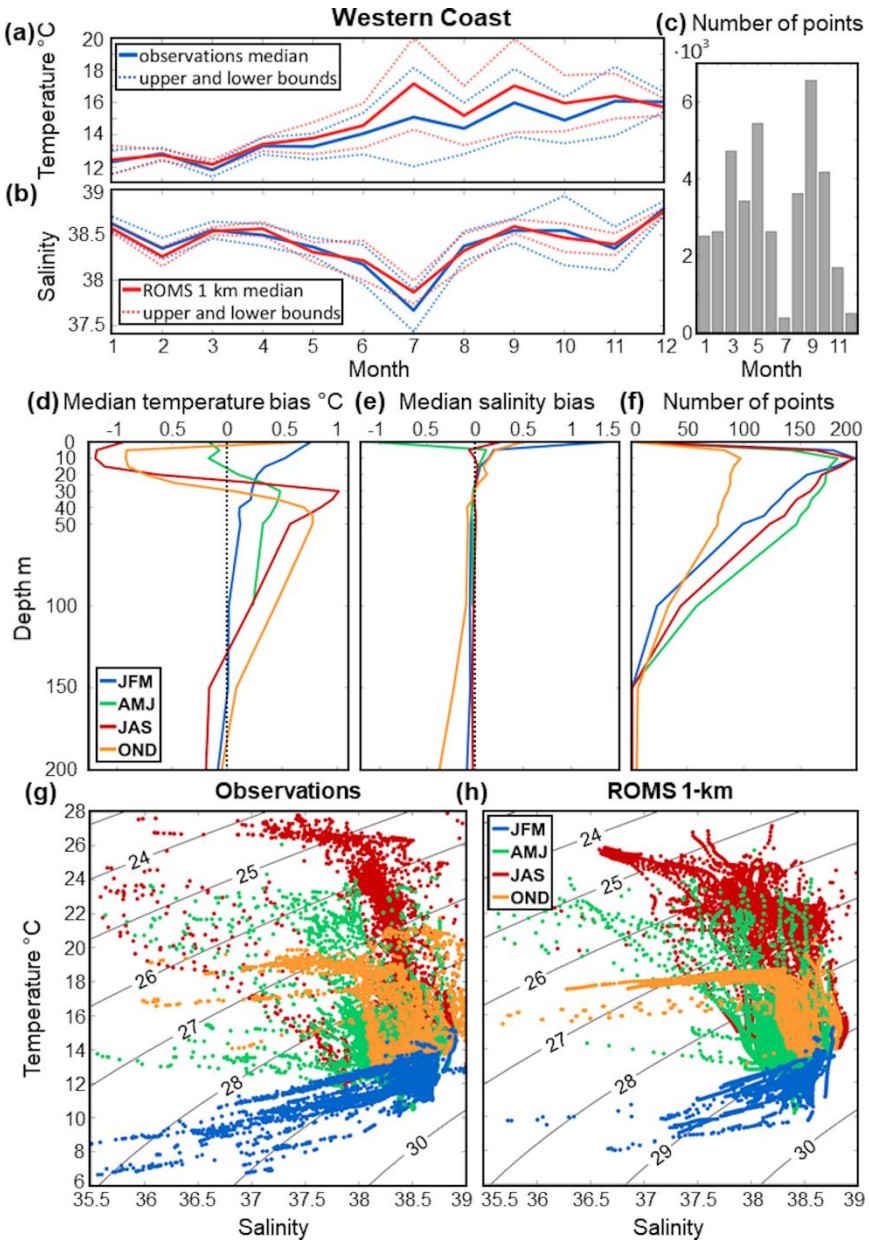

**Figure 9. Western Coast subdomain. Monthly climatology of AdriSC 1-km and *in situ* (a) median temperature, (b) median salinity and their variabilities (i.e. upper and lower bounds defined as ±MAD) as well as (c) number of observations per month. Seasonal variations of the (d) temperature and (e) salinity biases between the AdriSC ROMS 1-km model and observations depending on the depth as well as (f) number of observations per depth. Seasonal T-S diagrams for (g) the CTD observations and (h) the AdriSC ROMS 1-km model with PDA isolines.**



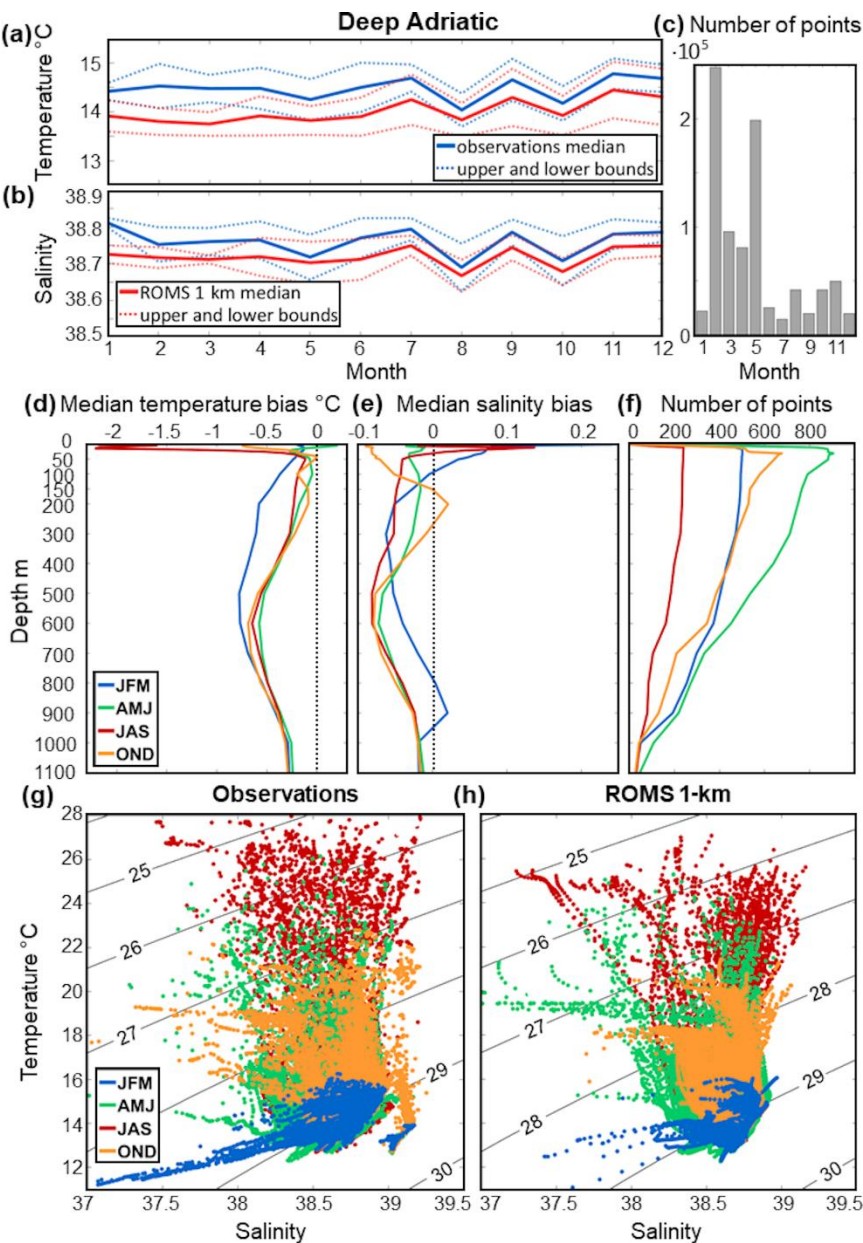

**Figure 10. Deep Adriatic subdomain. Monthly climatology of AdriSC 1-km and *in situ* (a) median temperature, (b) median salinity and their variabilities (i.e. upper and lower bounds defined as ±MAD) as well as (c) number of observations per month. Seasonal variations of the (d) temperature and (e) salinity biases between the AdriSC ROMS 1-km model and observations depending on the depth as well as (f) number of observations per depth. Seasonal T-S diagrams for (g) the CTD observations and (h) the AdriSC ROMS 1-km model with PDA isolines.**

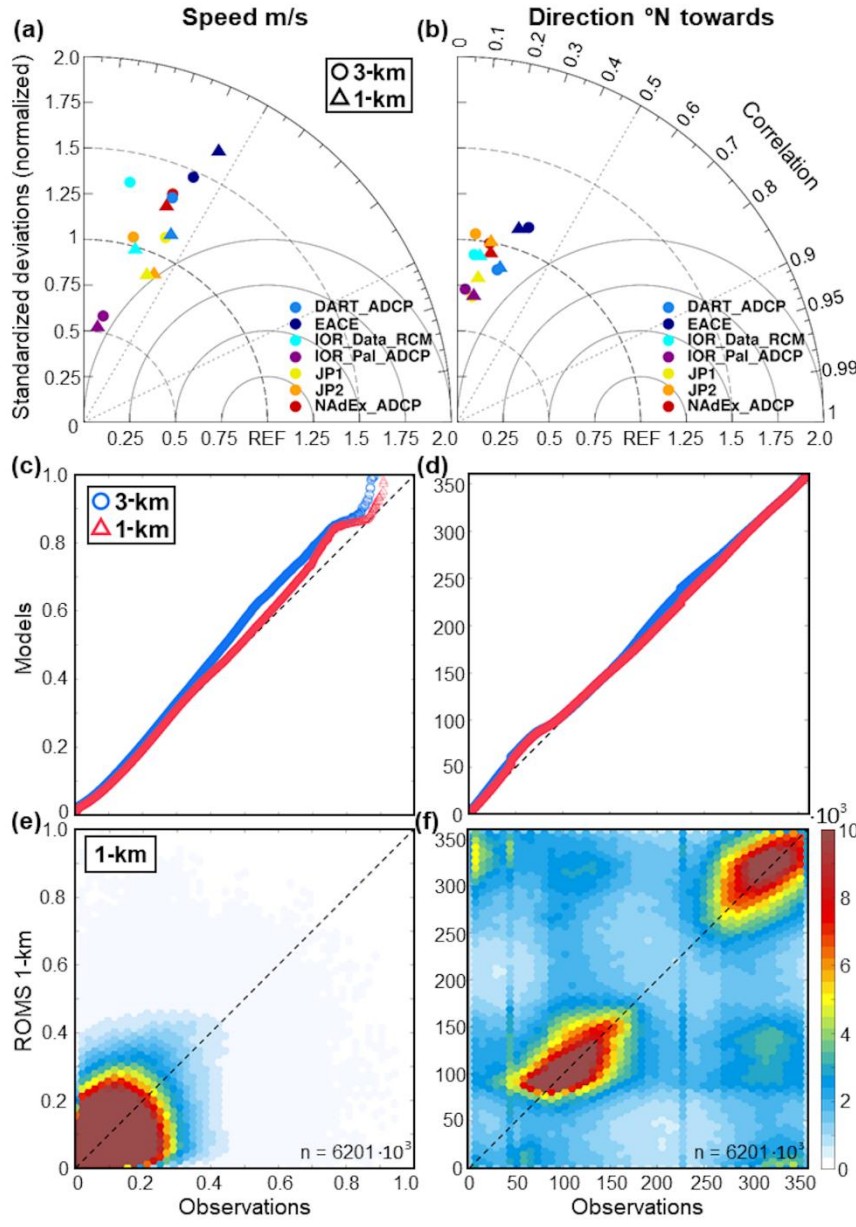

**Figure 11. Evaluation of the AdriSC ROMS 3-km and 1-km current speeds (left panels) and directions (right panels) against observations from 7 different datasets with (a, b) Taylor diagrams and (c, d) quantile–quantile plots as well as, only for the 1-km model, (e, f) scatter plots showing the density (number of occurrences) with hexagonal bins and total number of points n.**



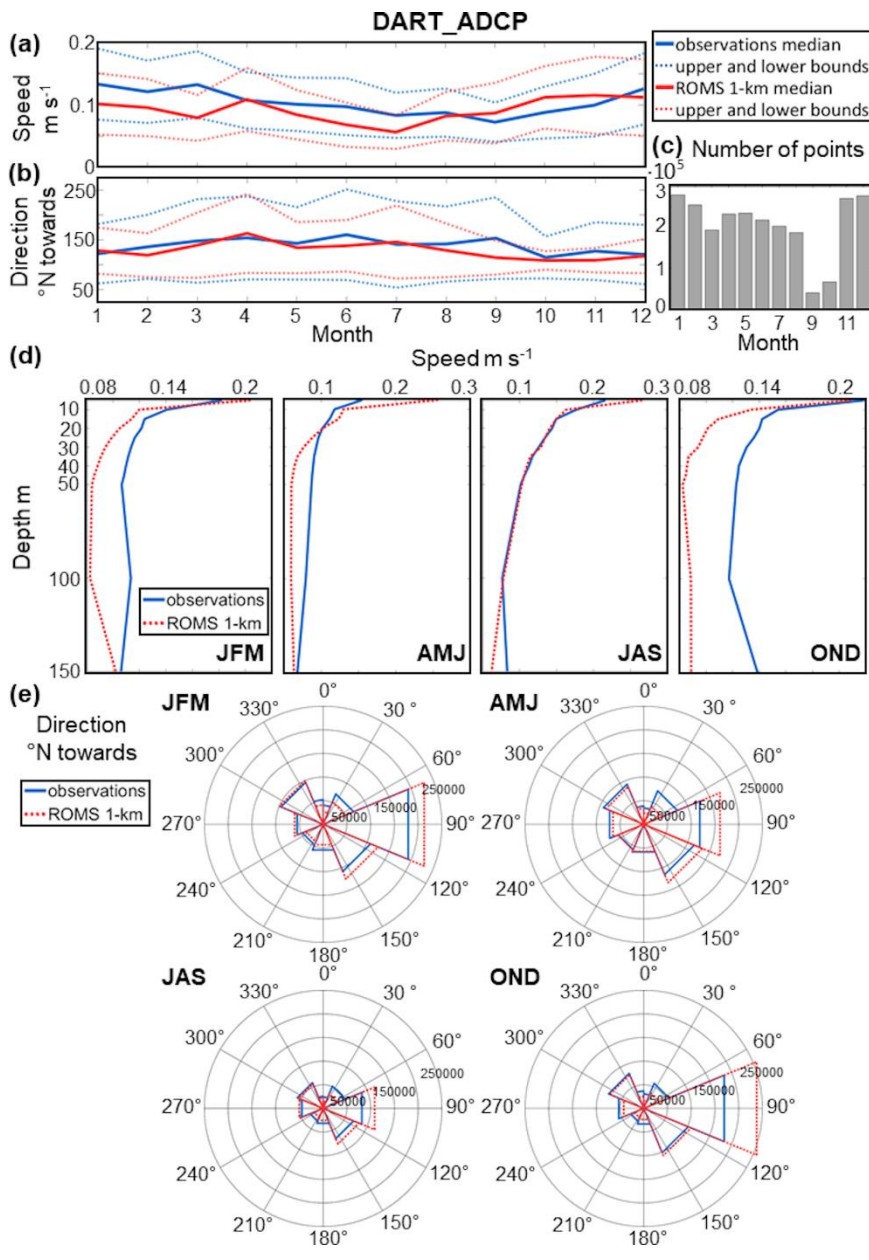

**Figure 12. DART_ADCP dataset. Monthly climatology of AdriSC 1-km and *in situ* (a) median speed, (b) median direction and their variabilities (i.e. upper and lower bounds defined as ±MAD) as well as (c) number of observations per month. Seasonal variations of the (d) speed of AdriSC ROMS 1-km model and observations depending on the depth. Seasonal rose plots of the (e) direction for ADCP observations and the AdriSC ROMS 1-km model.**

1000

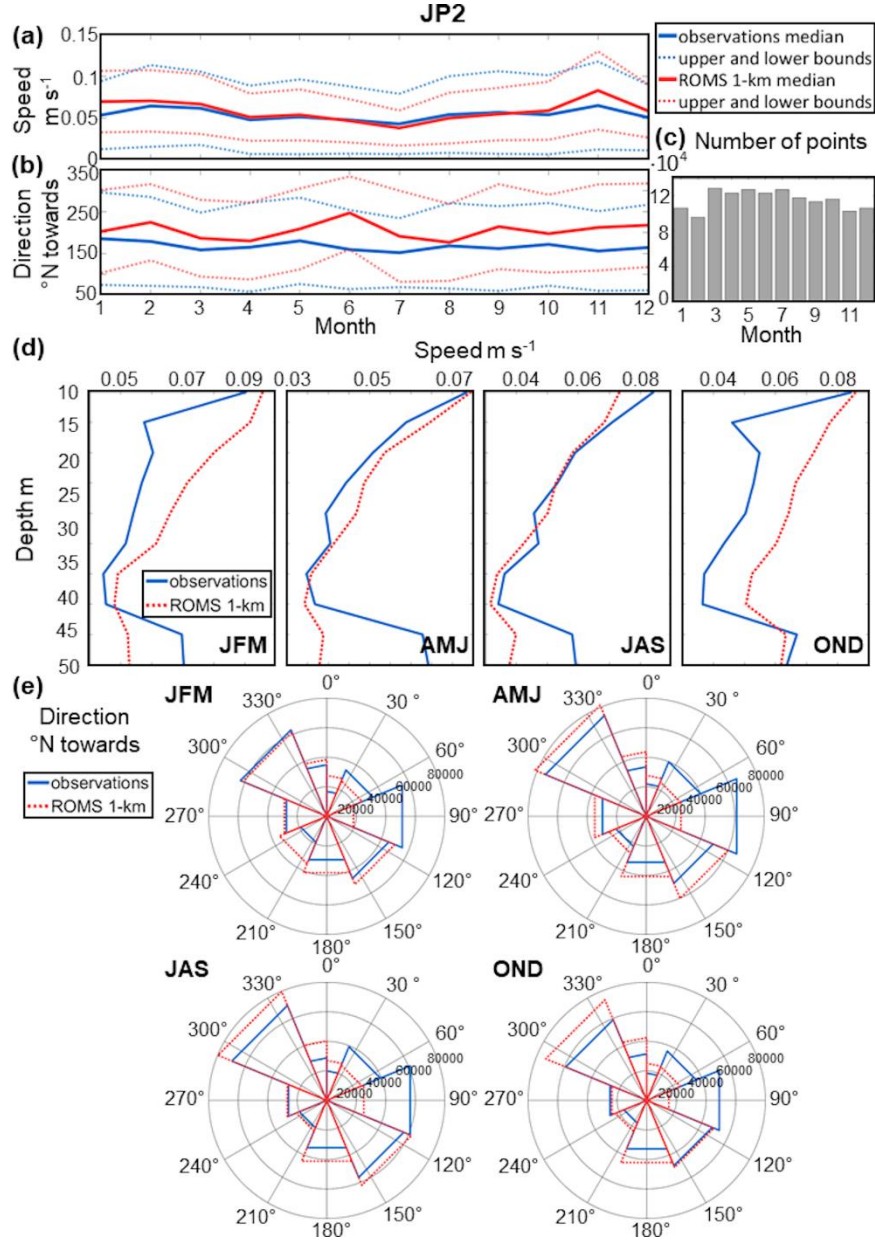

**Figure 13. JP2 dataset. Monthly climatology of AdriSC 1-km and *in situ* (a) median speed, (b) median direction and their variabilities (i.e. upper and lower bounds defined as ±MAD) as well as (c) number of observations per month. Seasonal variations of the (d) speed of AdriSC ROMS 1-km model and observations depending on the depth. Seasonal rose plots of the (e) direction for ADCP observations and the AdriSC ROMS 1-km model.**



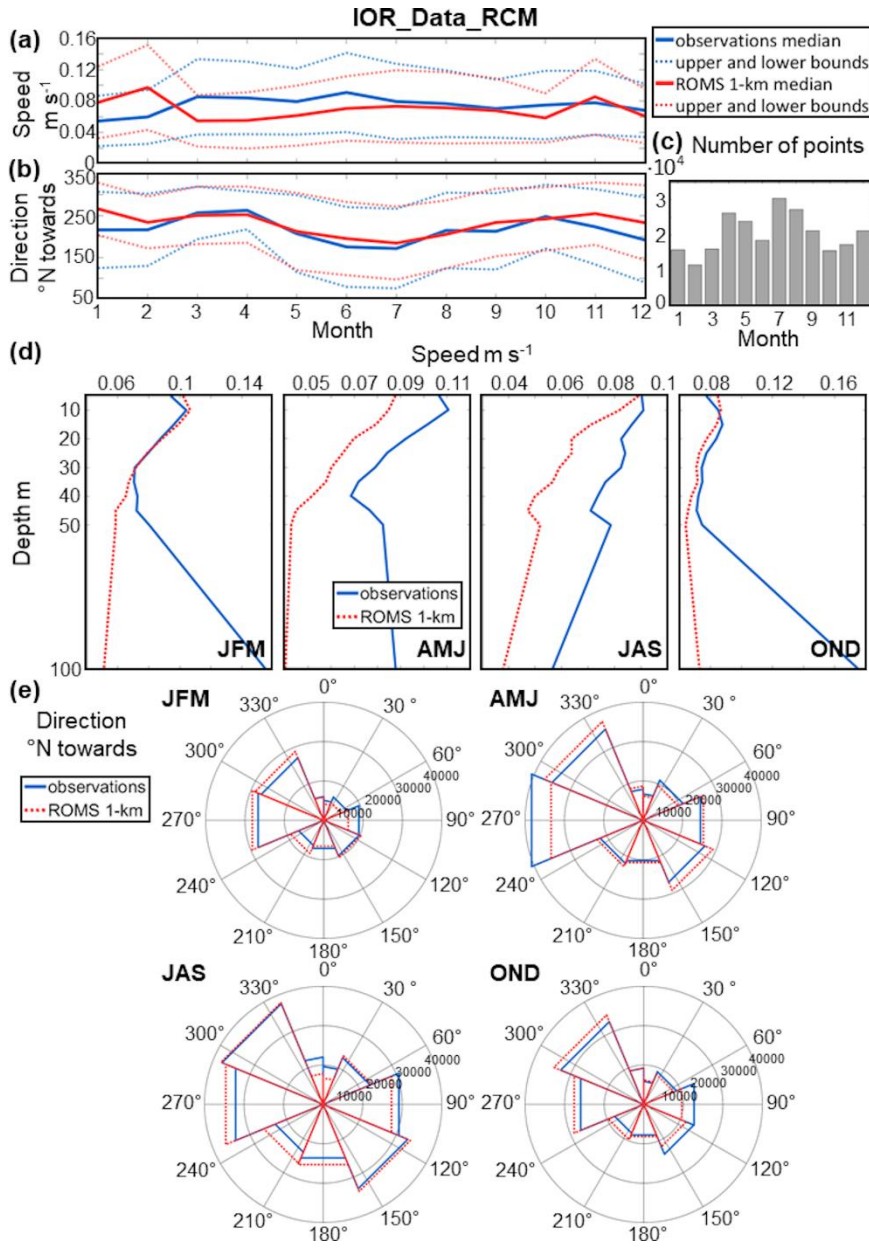

**Figure 14. IOR_Data_RCM dataset. Monthly climatology of AdriSC 1-km and *in situ* (a) median speed, (b) median direction and their variabilities (i.e. upper and lower bounds defined as ±MAD) as well as (c) number of observations per month. Seasonal variations of the (d) speed of AdriSC ROMS 1-km model and observations depending on the depth. Seasonal rose plots of the (e) direction for RCM observations and the AdriSC ROMS 1-km model.**



| Ocean model | ROMS | |
|---|---|---|
| Number of domains | 2 | |
| Horizontal resolution | 3 km | 1 km |
| Vertical resolution | 35 | |
| Time step | 150 s | 50 s |
| Atmospheric forcing (frequency) | WRF 3-km (30-min) | |
| Initial and boundary conditions (frequency) | MEDSEA (daily) | |
| 31-year period | 1987-2017 | |
| Frequency of outputs | Hourly | |

**Table 1. Summary of the AdriSC climate component ocean model main features for the evaluation run.**





| Dataset | Observations | Period | # locations | # records ($10^3$) | Max. depth (m) |
|---|---|---|---|---|---|
| SSHA | JPL MEASURES | 1992-2017 | 2065 | 3808 | surface |
| SST | AVHRR | 1987-2017 | 966 | 10938 | surface |
| SST | JPL MUR | 2002-2017 | 46777 | 266348 | surface |
| CTD | ARGO | 2012-2017 | 2182 | 569 | 1503 |
| CTD | ASCOP | 1990-1991 | 96 | 4 | 39 |
| CTD | CSP01 | 1991 | 108 | 5 | 64 |
| CTD | DART_CTD | 2005-2006 | 502 | 64 | 1202 |
| CTD | IOR_Data_CTD | 1987-2017 | 3043 | 419 | 1214 |
| CTD | IOR_Pal_CTD | 2012 | 5 | 4 | 170 |
| CTD | MEDATLAS | 1987-1990 | 254 | 63 | 2143 |
| CTD | NAdEx_CTD | 2014-2015 | 19 | 4.5 | 93 |
| CTD | OTRANTO | 1994-1995 | 332 | 231 | 1259 |
| CTD | PALMAS | 1994 | 103 | 14 | 1154 |
| CTD | PCO | 1987-1989 | 162 | 6 | 52 |
| CTD | POEM | 1991-1992 | 85 | 44 | 1191 |
| CTD | PR2_UR | 1996-1998 | 111 | 0.6 | 62 |
| CTD | PRISMA | 1995-1996 | 538 | 236 | 1208 |
| CTD | PRV | 1987-1988 | 283 | 1 | 38 |
| CTD | RB_NAd | 1987-2017 | 6 | 9 | 40 |
| CTD | SIRIAD_15 | 2015 | 64 | 26 | 1199 |
| CTD | All data | 1987-2017 | 7781 | 1700 | 2143 |
| ADCP/RCM | DART_ADCP | 2005-2006 | 11 | 2482 | 164 |
| ADCP/RCM | EACE | 2002-2003 | 2 | 282 | 68 |
| ADCP/RCM | IOR_Data_RCM | 1987-2004 | 321 | 268 | 930 |
| ADCP/RCM | IOR_Pal_ADCP | 2012 | 2 | 313 | 129 |
| ADCP/RCM | JP1 | 2007-2009 | 4 | 430 | 82 |
| ADCP/RCM | JP2 | 2013-2014 | 10 | 1784 | 79 |
| ADCP/RCM | NAdEx_ADCP | 2014-2015 | 8 | 940 | 83 |
| ADCP/RCM | All data | 1987-2015 | 358 | 9034 | 930 |

**Table 2. Name and period of the observations, number (#) of locations and records as well as maximum measured depth for the 4
1015 datasets – i.e. (1) Sea Surface Height Anomalies (SSHA), (2) Sea Surface Temperatures (SST), (3) Conductivity Temperature Depth
(CTD) observations and (4) Acoustic Doppler Current Profiler (ADCP) or Rotor Current Meter (RCM) measurements – used to
evaluate the AdriSC ROMS 3-km and 1-km models over the 1987-2017 period.**