# Peer review of "Performance of the Adriatic Sea and Coast (AdriSC) climate component – a COAWST V3.3-based one-way coupled atmosphere-ocean modelling suite: ocean results"

_Geoscientific Model Development, 2021_

## Author Response (AR1)

**Response to Reviewer #1 comments**

**General comments:**

*The paper of Pranic et al. entitled '*Performance of the Adriatic Sea and Coast (AdriSC) climate component – a COAWST V3.3-based coupled atmosphere-ocean modelling suite: ocean part*', presents an evaluation of the ocean component of the AdriSC climate system against a huge data collection. This study complete a previous evaluation paper that was dedicated to the atmospheric component of the same numerical model run of a 31-year long period (1987-2017). The ocean evaluation is conducted for sea surface, thermohaline properties and circulation.*

*My main concern, that I detail below, is about the numerical set-up. In particular I have difficulties to understand the ocean-atmosphere interface, as information about coupling/forcing are clearly missing. See also in the following for some questions about the ocean model itself.*

*The comparison to observations is fully described and very well presented, even if a small number of conclusions appears rapidly set. Also, some paragraphs are difficult to follow, when describing the subdomains results notably, but this appears inherent of the text insertion of the results put in the supplementary material.*

*That said, I suggest a minor revision will be useful to improve the paper before accepting its publication.*

**Response**: Thank you very much for your detailed and constructive review. In the other sections below, concerns about the model set-up have been appropriately addressed while the text has been changed in order to simplify the reading of the manuscript.

--- --- --- --- ---

**Main comments:**

**Coupling:**

*The first paragraph of section 2.1 seems to stand that WRF 3km and the ROMS models are coupled. In particular, the sentence "Finally, the data exchanges … are achieved with the Model Coupling Toolkit…". This needs some precision in my opinion.*

*Denamiel et al. 2021 introduced the fact that "the SST from ROMS grid is not prescribed to the WRF models". This means that WRF and ROMS executions are parallel but there is one-way interaction only? If so what are exactly the fields exchanged from WRF to ROMS?*

**Response**: The following sentence has been added to the paragraph in order to address the 2 above comments:
"Additionally, as no ROMS grid was set-up to entirely cover the spatial domain of the WRF 15-km grid, the ROMS sea surface temperature (SST) is not prescribed to the WRF models

in order to avoid the generation of discontinuities along the border between the two-way nested WRF 15-km and WRF 3-km atmospheric grids. Consequently, the only grid exchanges in the AdriSC modelling suite consist in the WRF 3-km model providing atmospheric fields (i.e. horizontal wind at 10 m, temperature at 2 m, relative humidity at 2 m, mean sea-level pressure, downward shortwave radiations, longwave radiations, rain and evaporation) to the ROMS 3-km and 1-km models which increases the efficiency of the AdriSC model."

*Do you plan to test the two-way coupled mode for future climate simulations? If yes, how will be managed the use of two ocean models? Do you think the dQ/dSST procedure will still be appropriate in the fully coupled run?*

**Response**:

For the RCP 8.5 scenario simulation following the PGW methodology soon to be completed, the one-way coupling has been kept identical in order to be able to compare the past and future simulations. This means that the MEDSEA SST is modified with a climatological change (e.g. increase of temperature up to 3.5 °C in summer) and imposed as boundary condition in the WRF grids as well as SST of reference in the dQ/dSST procedure. Consequently, (1) the WRF models do not benefit from the more accurate calculation of the future SST done in the ROMS models and (2) the tuning of the radiation fluxes is done with an approximated SST of reference. Unfortunately, this is the trade that the authors had to make in order to keep the high-resolution of the ocean models and to be able to conduct their research with their limited numerical resources.

Indeed, in the authors' opinion, as the Adriatic Sea truly needs to be described with at least a 1-km resolution (due to the complex coastline including many islands in its eastern side), the only way to achieve a full two-way coupling between WRF and ROMS models is to set-up a ROMS 9-km grid covering the same domain than the WRF 15-km grid. In this way the ROMS SST could be safely used as homogeneous (i.e. no discontinuities) boundary condition for the WRF grids. However, the AdriSC model is already extremely slow (1 month of results per day) and the addition of a new grid as well as new grid exchanges would definitely increase the computation time. As our modelling team is composed of only one expert numerical modeller and our numerical resources only consist on the use of the ECMWF supercomputing facilities via Special Project grants, it is unrealistic for us to run such a model.

The following paragraph has been added in the text after the previous addition already done above:
"Ideally, a two-way coupled system would require the use of an additional ROMS 9-km grid covering the WRF 15-km domain. However, due to limited numerical resources and the slowness of the AdriSC modelling suite, such a set-up could not be envisioned in this study. As a consequence, within the AdriSC modelling suite, the WRF models do not benefit from the more accurate simulation of the SST done with the ROMS models even for future scenario runs which only add climatological changes (e.g. increase of SST up to 3.5 °C in summer) to the SST forcing used in the evaluation run."

Concerning the dQ/dSST procedure, the authors think it could still be used in a fully coupled system as it just tempers with the solar radiation fluxes which are highly parameterized and not necessarily adjusted to the Adriatic Sea region. However, for scenario runs in the PGW methodology, this means that the SST of reference is just changed with a climatological modification while the real SST used as boundary condition in the WRF models is simulated with the ROMS models. This may thus have some unattended consequences in terms of fully coupled system and thus should be tested.

That said, the dQ/dSST should only be seen as a quick (dirty?) fix and not as a permanent solution. The authors believe that research on, and fine tuning of, the solar radiation penetration in the Adriatic Sea should be carried out in order to properly adjust the parametrizations in this region of the world. As suggested below by the reviewer, the use of ocean colour, turbidity and even the setup of a coupled sediment transport component are avenues that should definitely be explored in such research. However, as mentioned before, our research group is small and limited by its access to numerical resources. Consequently, the authors did not have the resources (human, financial, numerical) to carry out such experiments.

The following paragraph has been added in the text after the description of the dQ/dSST procedure:
"It should be noted that the use of the dQ/dSST procedure should not be seen as a permanent solution for climate studies in the Adriatic Sea. Indeed, the SST of reference used in future climate scenario runs is based either on other climate model predictions which are by nature uncertain or on approximations using climatological changes. Consequently, long-term research on the fine tuning and parametrization of the solar radiation penetration using, for example, ocean colour, turbidity or even sediment transport modelling, is thus a prerequisite to a better representation of the coupled atmosphere-ocean dynamics in the Adriatic Sea."

***Ocean model:***

*- I understand from run_coaswst_model.job0 in the package that the simulation start on 1st January 1987. This must be indicated in the text. Is there any spin-up of the ocean that affect the results for the first simulation years or the use of the MEDSEA re-analysis permits to rapidly have an equilibrium? Does the choice of starting in winter have an impact on dense water formation?*

**Response**: The authors fully agree with the reviewer that the description of the initial conditions and spin-up of the models is missing in the manuscript and the following paragraph has been added after the description of the dQ/dSST procedure:
"Fourth, the AdriSC evaluation run was initialized the 1$^{st}$ of November 1986 in order to have a short two-month spin-up period allowing the ocean models to reach a steady state. Indeed, short experiments have shown that rapid equilibrium is reached within the AdriSC ocean models due to (1) the use, before the 1$^{st}$ of January 1987, of monthly (instead of daily) MEDSEA v4.1 re-analysis products which have a relatively fine resolution (about 9-km) and assimilate all available data in the Mediterranean Sea and (2) the relatively small size of the ROMS ocean domains. Ideally, several long-term simulations should have been run with different spin-up periods in order to better quantify the impact of the initial

conditions on the long-term ocean model results. However, due to numerical resources limitations, such systematic tests have not been carried out with the AdriSC climate model."

*- l161: Are the river flows distributed homogeneously on the 20 first levels or is there any flow vertical profile?*

Response: A vertical profile is applied to distribute the river flow. The following sentence has been modified:
"Additionally, the river flows are linearly distributed between the 20 first sigma vertical levels – i.e.  the discharge is multiplied by weights ranging from 20/210 at the surface, 19/210 at the 1st sigma level below the surface, to zero at the 20th sigma level below the surface."

*- l162-165: It seems that it could be relevant for the Adriatic Sea to take the sea water colour and turbidity effect on the Fadd radiation penetration. This is also mentioned in the conclusion. Is there something in ROMS that can be tested or introduced in this direction? Or is there any interest to add the sediment component of COAWST?*

Response: see response in "main comments" section.

--- --- --- --- ---

**Other comments:**

*1. Introduction*

*- l59: It can be relevant to complete with Carniel et al. (2016)'s study which is more focusing on dense water formation during the 2012 Bora event.*

Response: Accepted, the reference has been added.

*2. Model, data and methods* (see my main comments)

*- l278: Fig. 1b → Fig. 1c (for the 7 subdomains)*

Response: Reference to figure 1c has been added.

*- l281-282: the sentence is cut*

Response: Thank you, the typo problem has been fixed.

*3. Results and Discussions*

*- l370-372: In my opinion, the results summary in section 3.1.2 should be separated in two sentences to be fair.*
*"... the model is capable to reproduce the BiOS, even though with a weaker intensity due to the overestimation of both seasonal and interannual signals..." "... the SST is quite well reproduced despite presenting a persistent cold bias within the Adriatic Sea."*

**Response**: Accepted. Sentence has been split.

*- l400 and 403: to avoid some confusion (like mine), I suggest to use "the CSP01 dataset" instead of "experiment".*

**Response**: Accepted. "experiment" has been replaced with "dataset"

*- Paragraphs from lines 501-516 and lines 606-616 are difficult to follow. This of course is related to the separation with the supplement material (that is a good option), but if possible, it would be better to find a clearer organization for these two parts either by describing region by region, or by separating temperature results from salinity results for the first paragraph.*

**Response**: The article has been revised in order to clarify and reorganize the text to describe the results region by region.

*- l519: "… the analysed subdomains and mostly with a good accuracy." For me, you should delete "and".*

**Response**: The sentence has been rewritten as follow:
"In summary, the evaluation of the AdriSC ROMS 1-km thermohaline properties shows that the model is overall capable to reproduce, with mostly a good accuracy, the temperature and salinity in all of the analysed subdomains."

*- l537: The fact that the highest biases are found around the thermocline and halocline is generally due too smoothed vertical gradients at the ocean mixed layer base. Is this what you obtain in the AdriSC ROMS models? If yes is it equivalent in the 1km-resolution than in the 3-km resolution? Is it somehow related to a similar default in the MEDSEA reanalyses that persists from initial condition or propagates through the open boundaries?*

**Response**: The authors indeed believe that the observed biases found around the thermocline and/or halocline can be attributed to smooth vertical gradients near the mixed layer base. However, to be fair, as the authors did not systematically tested different parametrizations for the vertical mixing and diffusivity in the ROMS models, they cannot discriminate whether the obtained biases are due to the ROMS model setup itself or the MEDSEA forcing. This is an interesting point that should definitely be investigated in further studies. The following paragraph has been included in the text:
"Additionally, independently of the subdomains, the analysis of the vertical profiles shows that the temperature and salinity biases often present a peak in the vicinity of the thermocline/halocline depth which can probably be linked to an inaccurate representation of vertical diffusivity and vertical mixing in the AdriSC ROMS models. However, more in-depth work should be done to discriminate whether the vertical biases are linked to the AdriSC ROMS model set-up per se or to the MEDSEA fields used to force the initial and boundary conditions."

*-l637-640: "This highlights… the hurricane strength bora winds." This conclusion about the role of wind variability appears quite rapidly set here. Maybe you should precise the*

*resolution of ALADIN? Did you make some tests using WRF 15km to drive the AdriSC ROMS models?*

*There is also a typo in "particular".*

**Response**: The authors forget to put the reference of their work concerning the impact of the resolution of the atmospheric models on the representation of both the bora strength and the sea-surface cooling (using ERA5, WRF 15-km, WRF 3-km and WRF 1.5-km). The paragraph has been modified as follow:
"This highlights that, in the north-eastern Adriatic, higher horizontal and vertical ocean and atmospheric model resolutions, better resolving the complex bathymetry and orography, are required to reproduce the mesoscale variability of the winds and particularly the hurricane strength bora winds as demonstrated by Denamiel et al. (2021a)."

**4. Conclusion**

*- l645: To be fair the word "coupled" may be changed. But yes to my knowledge too this is the first time despite several initiatives and still many challenges (see Schär et al. 2020 for instance)*

**Response**: Accepted. Coupled has been removed and the reference has been added as follow:

"In the presented study, the evaluation of the AdriSC ROMS 3-km and 1-km ocean models – forced by the already evaluated AdriSC WRF 3-km model (Denamiel et al., 2021b) – has been carried out for the 31-year long period (1987-2017). The main novelties of the work are, first, the implementation for the very first time – at least to the author's knowledge – of a kilometer scale atmosphere-ocean model for long-term climate studies which still present many challenges (Schär et al., 2020) and, second, the amount of in situ data collected to perform the evaluation of both daily thermohaline (CTD measurements) and hourly dynamical (ADCP and RCM observations) properties of the AdriSC ocean models."

**Response to Reviewer #2 comments**

*The authors developed a coupled atmosphere-ocean model system in the Adriatic Sea. They evaluated the ocean part performance of the coupled model in this manuscript (MS).*

*It is challenging to develop an ocean model in a dynamic region with complex bathymetry and applied it to a long-term simulation. In MS, the authors implemented the coupled model system for a 31-year simulation. Model simulated SSH, SST, temperature, salinity, as well as current, were validated by the satellite measurements and in-situ observations. Methods they chose for the validation, such as Taylor diagram, MAD, T-S diagrams and so on, are widely used in skill assessment. As the result, the model can reproduce dynamical properties and the general pattern of the variations.*

Response: Thank you very much for your review.

Here are comments and suggestions:

1. *Line 90: Liu et al. (2021) wasn't listed in the "References".*

Response: Accepted. It has been added to the manuscript.

2. *Section 2.1: Is the nesting 2-way or 1-way?*

Response: The ocean grids are one way nested. The sentence in the text was modified as follow:
"… (2) the complex coastal Adriatic Sea dynamics with a one-way nested 1-km grid (676 x 730)."

3. *Section 3.1.1: Looks like there is a conspicuous difference in the EOF1 amplitude. Did the authors calculate correlation coefficients between the amplitude of observation and amplitude of the model?*

[Figure]

Figure R1: Time variations of the amplitude of EOF1 for both observation and model.

**Response**: The correlation coefficient between observed and modelled EOF1 is 0.65 and the normalized standardized deviation is 1.19 which shows that the model amplifies the seasonal signal by comparison to the observations (see Fig. R1 above). The manuscript has been changed as follow:
"Overall, it can clearly be seen that, for both Adriatic and northern Ionian Seas (Fig. 2), the first EOF component (EOF1) represents the seasonal variability of both AdriSC ROMS 3-km and JPL_MEASURES results with spatial signal and amplitudes slightly stronger in the model (i.e. 81.2% of the total signal with amplitudes varying between ±8.0; Fig. 3) than in the observations (i.e. 74.5% of the total signal with amplitudes ranging between ±6.0; Fig. 3). Additionally, the correlation coefficient between the time variations of the observed and modelled EOF1 is only 0.65 associated with a normalized standard deviation of 1.19."

4. *Section 3.1.2: Comparing to the reference, the standard deviation of salinity is quite low (~0.25, Figure 6b). This should be mentioned in MS.*

**Response**: The authors think the reviewer means Section 3.2.1 and not 3.1.2 which only discusses the SST and SSH variabilities. Additionally, the misrepresentation of the salinity is already discussed in the manuscript in section 3.2.1:
"… but do not properly capture the observed salinity (i.e. correlations around 0.7 and normalized standardized deviations between 0.3 and 0.5)."

5. *Section 3.2.1: Median temperature bias reaches almost four degrees in the subsurface (30m, Figure 8d). Can the authors explain why the model has such a large bias in the subsurface?*

**Response**: The authors believe that the observed biases found around the thermocline and/or halocline can be attributed to smooth vertical gradients near the mixed layer base. However, to be fair, as the authors did not systematically tested different parametrizations for the vertical mixing and diffusivity in the ROMS models, they cannot discriminate whether the obtained biases are due to the ROMS model setup itself or the MEDSEA forcing. It should also be noted that for this specific case, the number of measurements is largely decreased where the extreme bias occurs. This means that, if unrealistic measurements were not flagged during the quality check process, they may weight in far much than when more observations are available. The following paragraph has been included in the text:
"Additionally, independently of the subdomains, the analysis of the vertical profiles shows that the temperature and salinity biases often present a peak in the vicinity of the thermocline/halocline depth which can probably be linked to an inaccurate representation of vertical diffusivity and vertical mixing in the AdriSC ROMS models. However, more in-depth work should be done to discriminate whether the vertical biases are linked to the AdriSC ROMS model set-up per se or to the MEDSEA fields used to force the initial and boundary conditions."

6. *Section 3.3.1: The correlation coefficient of direction looks very poor in the Taylor diagram. However, in the Q-Q plot, the modeled direction matches observation very well. Can the author explain it?*

**Response**: The difference between Q-Q plot and correlation coefficient can easily be explained by the fact that Q-Q plots compare the distributions of the variable not their value at each time like with the correlation coefficient or the scatter plot. Generally speaking, it is

good to keep in mind that it is easier to obtain a matching Q-Q plot than for example a matching scatter plot as illustrated in Figure 11 of the article. In this case the authors attribute the time-phase differences to the tidal representation and the fact that the time of the archived data was not necessarily provided in UTC:

"However, due to the already mentioned lack of synchronization, modelled current speeds and most especially modelled current directions can be extremely spread compared to the observations. Despite the inherent difficulties to reproduce the ocean dynamics at the hourly scale, the scattering of the AdriSC ROMS 1-km results can also result from the uncertainties linked to the observational dataset time references. Indeed, due to the lack of metadata availability for a certain number of datasets, some observations which may have been provided in local time have been compared with model results in Universal Time Coordinated (UTC)."

---

## Author Response (AR2)

**Response to the Topical Editor review**

Dear Editor,

Thank you very much for your revisions which have been adressed in the sections below.

1- Why did you change "part" for "dataset" in the manuscript title? I think it is inappropriate as "dataset" can also refer to observations. I propose to change "dataset" for "analysis" or "results".

**Response:** The change was done during typesetting for the previous article dealing with the evaluation of the atmospheric part: https://gmd.copernicus.org/articles/14/3995/2021/
As the authors are not native english speakers they believe it is better to follow recommendations from typesetting experts. Consequently, to be homogeneous, the authors also changed the title of this article. Nevertheless, the title has been changed to "ocean results".

2- Regarding your reply to reviewer #1 main comment on "Coupling", I consider that it is relevant. However, I think that still more emphasis should be put on this aspect of one-way coupling:
- "one-way" should appear in the manuscript title : " Performance of the Adriatic Sea and Coast (AdriSC) climate component – a COAWST V3.3-based one-way coupled atmosphere-ocean modelling suite ..."

**Response:** "one-way" was added to the title.

- In section 4, the fact that your system is only one-way coupled should be repeated. e.g. by changing the 2nd sentence for "The main … of a kilometre-scale one-way coupled atmosphere-ocean model for long-term ..."

**Response:** "one-way" has been added to the 2nd sentence.

- In the paragraph starting at line 160, I think that you should emphasise that the boundary forcings you describe are needed because your system is not coupled in the ocean-to-atmosphere direction.

**Response:** Actually, paragraph originally starting at line 160 describes the boundary and initial conditions use to run the ROMS 3-km ocean model ... Consequently it is not related to the "one-way coupling" with the atmosphere. However, the authors added the following sentence after the description of the MEDSEA re-analysis in order to remind the reader that SST used in WRF is also coming from MEDSEA:
"It should be noted that the SST used in the WRF models is also provided by the MEDSEA re-analysis as fully described in Denamiel et al. (2021b)."

- In this paragraph, you should also describe in more detail how do you derive the future SST forcing for WRF; currently, the only thing you write about this is half of a sentence at lines 139-

140 and I consider this is not emphasized enough.

**Response:** The description of the PGW methodology developed by the authors for the ocean models is fully presented in their previous work: Denamiel et al. (2020a) and is not the objective of this evaluation study which only deals with the historical period not the projected climate changes. The authors thus propose to modify the paragraph starting at Line 136 as follow: "... As a consequence, within the AdriSC modelling suite, the WRF models do not benefit from the more accurate simulation of the SST done with the ROMS models. This is also true for future scenario runs which only add climatological changes (e.g. increase of SST up to 3.5 °C in summer) to the SST forcing used in the evaluation run following the Pseudo-Global Warming (PGW) method originally developed for the atmosphere (Schär et al., 1996) and extended to the ocean by Denamiel et al. (2020a)."

- Also in your conclusion, you should at least briefly discuss the limits of this aspect of one-way coupling and what benefit would be added with 2-way coupling. Reviewer #1 seems to ask for that type of discussion (see his/her last sentence in his/her comment "Coupling").

**Response:** The authors agree that the benefice of two-way coupling should be discussed. However, it is related to the atmospheric models and not to the ocean models which are the focus of this article. Consequently the authors added this sentence at Line 136 where the two-way coupling is already discussed:

"Ideally, a two-way coupling which imposes the SST of the ocean models to the atmospheric models should be used in climate studies. Indeed, it allows for better representation of the SST which is known to impact the local and regional precipitations (Mejia et al., 2018; Yang et al., 2019; Johnson et al., 2020). In the AdriSC modelling suite the two-way coupling would require the use of an additional ROMS 9-km grid covering the WRF 15-km domain. However, due to limited numerical resources and the slowness of the AdriSC modelling suite, such a set-up could not be envisioned in this study. "

3- To answer reviewer #1 comment on the ocean model, you reply that you added a whole paragraph on the initialisation and the spin-up "Fourth, the AdriSC evaluation … carried out with the AdriSC climate model." but I cannot find it in the revised version of the manuscript. Please add it!

**Response:** The authors apologize for not including the paragraph and have added it in the new version.

4- I understand that the 1-km ROMS is nested in the 3-km ROMS. This is clearly stated only once, at line 126. Can you repeat this at places in the text? Can you be more precise on how the one-way nesting works? What fields are transferred from the 3-km to the 1-km ? Table 1 should give more detail on that.
In general, I think you should avoid refering to 2 models but instead refer to one ROMS model

with a 1-km region nested in a larger 3-km region.

**Response:** "the one-way nested (AdriSC) ROMS 1-km" as been added at Lines 156, 218, 273, 280, 587, 683. To clarify the one-way nesting between ROMS 3-km and ROMS 1-km grid the following text has been added " ... with a one-way nested 1-km grid (676 x 730) receiving temperature, salinity, ocean currents and sea-surface elevation at its boundaries from the AdriSC ROMS 3-km model." Table 1 has been modify to include the fields exchanged in the one-way nesting.
The authors would like to keep the denomination "model" as the two grids receive different forcing at their boundaries and are not two-way coupled. In other words, the ROMS 3-km model downscale the MEDSEA re-analysis while the ROMS 1-km model downscale the ROMS 3-km model results.

5- l.567: I think it is improper to write that MEDSEA fields are used to force the initial and boundary conditions; MEDSEA fields are used AS initial and boundary conditions; can you correct the sentence?

**Response:** The sentence has been corrected.